# Patched regulates lipid homeostasis by controlling cellular cholesterol levels

Carla E. Cadena del Castillo [1], J. Thomas Hannich [2,6], Andres Kaech[3], Hirohisa Chiyoda[4], Jonathan Brewer[5], Masamitsu Fukuyama[4], Nils J. Færgeman [5], Howard Riezman [2] & Anne Spang [1✉]

Hedgehog (Hh) signaling is essential during development and in organ physiology. In the canonical pathway, Hh binding to Patched (PTCH) relieves the inhibition of Smoothened (SMO). Yet, PTCH may also perform SMO-independent functions. While the PTCH homolog PTC-3 is essential in *C. elegans*, worms lack SMO, providing an excellent model to probe non-canonical PTCH function. Here, we show that PTC-3 is a cholesterol transporter. *ptc-3(RNAi)* leads to accumulation of intracellular cholesterol and defects in ER structure and lipid droplet formation. These phenotypes were accompanied by a reduction in acyl chain (FA) length and desaturation. *ptc-3(RNAi)*-induced lethality, fat content and ER morphology defects were rescued by reducing dietary cholesterol. We provide evidence that cholesterol accumulation modulates the function of nuclear hormone receptors such as of the PPARα homolog NHR-49 and NHR-181, and affects FA composition. Our data uncover a role for PTCH in organelle structure maintenance and fat metabolism.

[1] Biozentrum, University of Basel, Basel, Switzerland. [2] Department of Biochemistry and NCCR Chemical Biology, University of Geneva, Geneva, Switzerland. [3] Center for Microscopy and Image Analysis, University of Zurich, Zurich, Switzerland. [4] Laboratory of Physiological Chemistry, Graduate School of Pharmaceutical Sciences, University of Tokyo, Tokyo, Japan. [5] Department of Biochemistry and Molecular Biology, Villum Center for Bioanalytical Sciences, University of Southern Denmark, Odense, Denmark. [6] Present address: CeMM - Research Center for Molecular Medicine of the Austrian Academy of Sciences, Vienna, Austria. ✉email: anne.spang@unibas.ch

The Hedgehog (Hh) signaling pathway is crucial during animal development and has also demonstrated roles independent of development stages. The Hh receptor PTCH is among the most mutated tumor suppressors[1] and more specifically, PTCH1 mutations are the cause of the Gorlin Syndrome[2]. In the classical Hh signaling pathway, PTCH inhibits the plasma membrane G-protein coupled receptor (GPCR) smoothened (SMO). Upon Hh binding to PTCH, this inhibition is relieved, and SMO can activate a downstream signaling cascade. The mechanism by which PTCH inhibits SMO was enigmatic for a long time because PTCH represses SMO without direct contact[3]. PTCH1 was shown to be able to transport cholesterol[4–6] which in turn will directly activate SMO[7], a finding, which was supported by recent structural analyses[1,8–10]. The structures suggest that Hh inhibits PTCH transporter function and hence plasma membrane cholesterol levels could increase. Such an increase of cholesterol might be sensed through the sterol sensing domain in SMO and thereby activate the GPCR. As PTCH may mainly function as a cholesterol transporter, it might also affect other signaling pathways. In fact, in recent years SMO-independent PTCH signaling has been reported[11–14]. However, the mechanistic understanding of these non-canonical Hh signaling pathways remains largely unknown.

*Caenorhabditis elegans* expresses two PTCH homologs, PTC-1 and PTC-3, which are essential for development and survival[15–17]. While PTC-1 function appears to be mostly restricted to the germline, PTC-3 is expressed in somatic tissues[18–20]. No clear SMO homolog is encoded in the genome. In addition, some of the other downstream targets of the canonical Hh signaling pathway are also missing. In fact, it was proposed that SMO and those components were specifically lost during evolution in nematodes[15,21–23]. For example, SUFU is not conserved and the homolog of the transcription factor Gli, TRA-1, is involved in sex determination and gonad development in males and hermaphrodites[24]. Therefore, *C. elegans* provides an excellent model to study non-canonical, SMO-independent Hh signaling pathways, in particular in somatic tissues. To dissect SMO-independent PTCH functions, we concentrated on PTC-3, which is expressed in somatic tissues, in particular in the hypodermis, glia, and gut[20].

Here, we show that reduction of PTC-3 levels causes the accumulation of intracellular cholesterol and reduction in polyunsaturated fatty acids (PUFAs). Moreover, the endoplasmic reticulum (ER) lost most of its reticulate tubular form and developed elaborate sheet structures in the intestine. This effect in turn strongly impaired lipid droplet biogenesis, resulting in the inability of the animal to store fat. Reduction of dietary cholesterol rescued fat content defects, the ER morphology defects, and improved development and survival in *ptc-3(RNAi)* animals. Cholesterol levels influence nuclear hormone receptor activity such as of the PPARα homolog NHR-49, which is involved in the regulation of FA synthesis. Thus, our data demonstrate that PTCH also controls intracellular cholesterol levels in *C. elegans*, Moreover, we show that PTCH thereby impinges on FA metabolism, organellar structure, and fat storage capacity.

## Results

**PTC-3 has cell-autonomous and non-autonomous functions and is required for lipid storage in the intestine**. In order to understand the function of PTCH proteins in *C. elegans*, we decided first to revisit the phenotypes caused by the depletion of the somatically expressed PTC-3. Like its mammalian homolog, PTC-3 is essential for development. Consistently, it has been reported that *ptc-3(RNAi)* results in growth, molting, and vulva morphogenesis defects[17,18]. Given the essential role of PTCH in

development, we started the knockdown by RNAi only at the L2 stage of development, allowing the worms to progress further in development and even some to reach adulthood. In addition to the previously reported phenotypes, we noticed that the *ptc-3 (RNAi)* animals were much paler than their mock-treated counterparts (Fig. 1A). Pale worms are an indication for defects in fat content. *C. elegans* has a much simpler body plan than humans and hence some *C. elegans* organs take over more functions. For example, the worm intestine has paracrine functions, and also serves as the fat storage organ[25]. Thus, in a simplified view, the *C. elegans* intestine represents the functional equivalent of the human intestine, adipose tissue, and liver. To test whether PTC-3 was expressed in the gut as indicated by genome-wide expression analyses[20], we raised antibodies against PTC-3 (Fig. S1A). Those antibodies decorated the apical membrane of gut epithelial cells, while no plasma membrane signal was detected in oocytes, consistent with the notion that PTC-3 is present only in somatic tissues (Fig. 1B). This localization was confirmed with a GFP-tagged PTC-3 (Fig. S1B).

To determine which phenotype is dependent on intestinal PTC-3, we performed a gut-specific knockdown of PTC-3[26]. *ptc-3 (RNAi^{gut})* animals were still paler and thinner than mock-treated animals (Fig. 1C). Moreover, vulva morphogenesis defects were also observed upon the *ptc-3(RNAi^{gut})* regime, indicating that PTC-3 has cell-autonomous and non-autonomous functions. In order to exclude any leakiness of the gut-specific RNAi, we knocked down POS-1, which is essential for embryonic patterning and specification[27]. While in N2 animals, no alive progeny could be obtained ($n = 28$ animals), in the intestinal-specific RNAi condition all animals gave rise to viable progeny ($n = 28$ animals) (Fig. S1C). Therefore, we conclude that the gut-specific RNAi is not leaky and that PTC-3 has also cell non-autonomous functions.

As outlined above, pale phenotypes are often associated with lipid storage defects in worms[25,28]. Nile Red staining indeed showed a reduction in lipid content in *ptc-3(RNAi)* animals (Fig. 1D, E). A drawback of Nile Red staining is that autofluorescence in the intestine caused by lysosome-related organelles (LROs) is also potentially measured at the same time, which may confound the results. Therefore, we turned to coherent anti-stokes Raman scattering microscopy (CARS), a dye-free method recognized for accurate in vivo lipid detection in worms[29]. This analysis confirmed the Nile Red staining, and we observed an about 50% reduction in lipid content (Fig. 1F, G), indicating that we can use Nile Red for further analysis. The CARS signal did not overlap with the autofluorescence of LROs (Fig. S1D). We conclude that loss of PTC-3 causes a reduction in fat content in the intestine. The defects we observe could be in general lipid content or more specifically in lipid storage. To distinguish between these possibilities, we measured the yolk content in oocytes. The yolk is produced in and secreted by the intestine and then endocytosed during oogenesis[30]. We did not find any significant difference in yolk uptake between mock and *ptc-3(RNAi)* animals (Fig. S1E, F). Thus, it seems likely that *ptc-3 (RNAi)* animals lose the ability to accumulate fat in the intestine.

**PTC-3 is a cholesterol transporter**
Recent data suggested that mammalian PTCH1 acts as a cholesterol transporter[1,5,6,8–10]. To investigate, whether PTC-3 shares the function of PTCH1 as cholesterol transporter, we first expressed PTC-3 in *Saccharomyces cerevisiae*, which does not contain any cholesterol[31] and measured cholesterol efflux from cells using TopFluor cholesterol in a pulse-chase experiment (Fig. 2A). PTC-3 expressing yeast cells exported cholesterol significantly faster out of the cell than control cells, similar to what

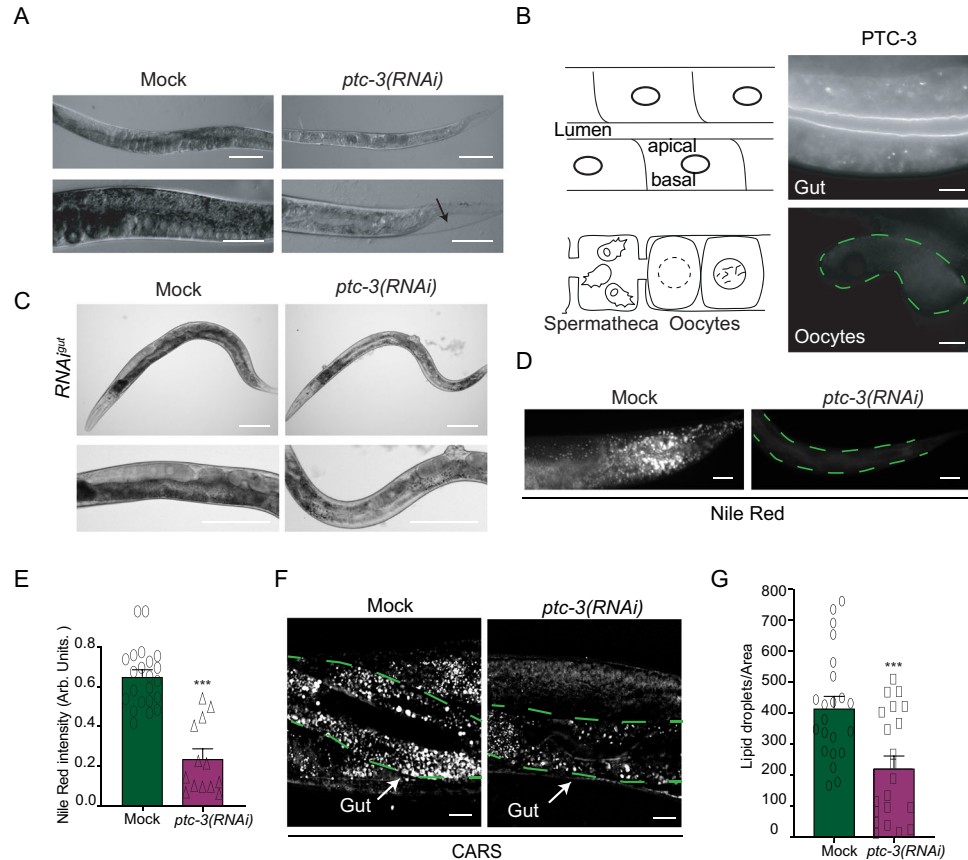

**Fig. 1 Loss of PTC-3 causes developmental defects and loss of fat content. A** Light microscopy images of adult N2 worms grown from L2 larva on mock or *ptc-3(RNAi)* bacteria. This feeding scheme was used for comparison between mock-treated and *ptc-3(RNAi)* adult animals. *ptc-3(RNAi)* animals are smaller, paler, and show cuticle defects. The arrow points to a cuticular defect. Scale bars upper panel 20 μm, lower panels 100 μm. n = 25 (mock) and 26 (*ptc-3(RNAi)*) animals from 3 independent experiments. **B** Immunofluorescence of isolated intestine and gonad from wild-type worms. Schematic representation of the intestine and the proximal gonad. PTC-3 is present at the intestinal apical membrane. n = 20 from 3 independent experiments. Scale bar 10 μm. (**C**) PTC-3 has cell-autonomous and non-cell-autonomous functions. Light microscopy images of adult gut-specific RNAi worms (*RNAi^gut*) grown from L1 larva on mock or *ptc-3(RNAi)* bacteria. *ptc-3(RNAi^gut)* animals show intestinal and vulval defects. Scale bars 100 μm. n = 16 (mock) and 27 (*ptc-3(RNAi)*) animals from three independent experiments. **D** Lipid content is reduced in *ptc-3(RNAi)* animals. Nile Red staining of adult mock and *ptc-3 (RNAi)* treated worms after L2 stage. Scale bars 20 μm. **E** Quantification of Nile Red staining shown in (**D**). Mean and SEM error bars. *** p = 0.000003, n = 22 (mock) and 13 (*ptc-3(RNAi)*) animals. One-way ANOVA including data on Fig. 3C. **F** CARS microscopy reveals reduction of lipid levels in live adult *ptc-3(RNAi)* after L2 stage fed animals. Scale bars 10 μm. The intestine is outlined by green dashed lines. Arrows point to the intestine. **G** Quantification of CARS signal. Mean and SEM error bars. ***p = 0.000524. n = 22 (mock) and 21 (*ptc-3(RNAi)*); one-way ANOVA. Statistical source data are available as source data.

has been observed for mammalian PTCH1[5]. To corroborate this finding, we expressed human PTCH1 and *C. elegans* PTC-3 in HEK293 cells and repeated the cholesterol efflux assays. Both PTCH1 and PTC-3 are able to promote cholesterol efflux from mammalian cells (Fig. S2A). This efflux capacity was dependent on an active permease domain, since a mutation in the permease domain[18], *ptc-3^{D697A}*, strongly reduced the cholesterol efflux (Fig. 2A). The *ptc-3^{D697A}* mutation has been reported to cause larval lethality in worms[18], establishing that cholesterol efflux is the essential function of PTCH. Next, we repeated the pulse-chase experiment in worms. While in mock-treated worms, TopFluor cholesterol was present mostly in the gut lumen, it was still strongly accumulated in the intestine in *ptc-3(RNAi)* worms after the washout, further demonstrating the role as cholesterol transporter (Fig. 2B). Finally, we measured sterol levels by mass spectrometry. Cholesterol levels were increased in *ptc-3(RNAi)* worms, while 7-dihydrocholesterol (7-DHC) and lophenol levels were decreased (Fig. 2C). 7-DHC and lophenol are downstream products of cholesterol in worms, indicating that cholesterol metabolism might also be affected by *ptc-3(RNAi)*. Taken

together our data strongly suggest that PTC-3, like PTCH1, is a cholesterol transporter at the plasma membrane.

**Cholesterol accumulates predominantly in the apical membrane in the intestine of *ptc-3(RNAi)* animals.** Cholesterol accumulates in *ptc-3(RNAi)* worms because it cannot be pumped out of the cells. In addition, cholesterol is not efficiently metabolized into 7-DHC and lophenol under those conditions. We speculated where the excess of cholesterol would reside in the cell. First, we used filipin, which binds specifically to cholesterol. While we could barely detect any filipin staining in mock-treated animals, *ptc-3(RNAi)* worms showed a strong fluorescent signal in the apical membrane in the intestine and also some appreciable increase in intracellular fluorescence (Fig. 2D). To corroborate this finding, we next employed two versions of the domain 4 of perfringolysin fused to mCherry probe (D4-mCherry), YDA and YQDA, which have different sensitivities in the detection of cholesterol[32–34]. We expressed the probes constitutively in the *C. elegans* intestine and analyzed their cellular distribution. Similar to the filipin staining, the mCherry signal increased in the plasma

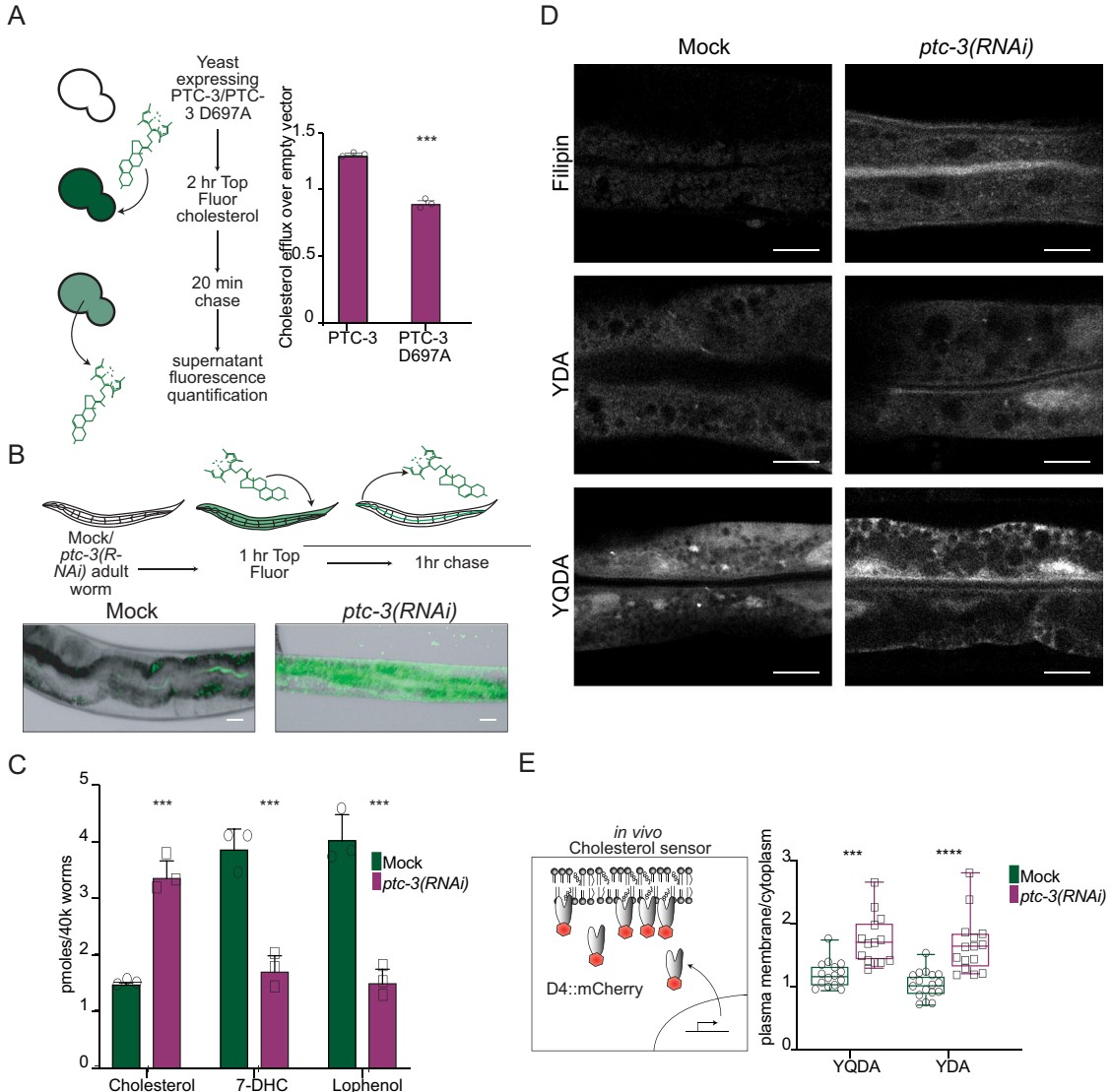

**Fig. 2 PTC-3 is a cholesterol transporter. A** *S. cerevisiae* expressing PTC-3 or PTC-3[D697A] were incubated for 2 h with 5 mM TopFluor® Cholesterol, washed, re-suspended in cholesterol-free buffer, and after 20 min, fluorescence intensity of the supernatant was measured. PTC-3 expression induced cholesterol efflux from yeast, which was abolished by the D697A mutation. Mean and SD error bars. Unpaired two-tailed $T$ test ***$p = 0.000066$ $n = 3$. **B** N2 adult worms fed with mock or *ptc-3(RNAi)* after L2 stage were incubated with 2.5 mM TopFluor® Cholesterol for 1 h. After 1 h chase, animals were imaged. Scale bar 10 μm. $n = 12$ (mock) and 16 (*ptc-3(RNAi)*) animals from three independent experiments. **C** Cholesterol accumulates in animals fed with *ptc-3(RNAi)* after L2 stage. Quantification of sterols by MS. Mean and SEM error bars $n = 3$. One-way ANOVA ***$p = 0.0003$. **D** Cholesterol identification by Filipin or mutagenized YDA or YQDA D4::mCherry cholesterol sensor in the adult intestine in mock or *ptc-3(RNAi)* treated animals after L2 stage. *ptc-3 (RNAi)* induces membranal cholesterol accumulation in the intestinal apical membrane. Scale bars 10 μm. **E** Quantification of apical membrane enrichment over cytoplasm of the cholesterol sensors YDA or YQDA D4::mCherry. Whiskers represent min and max, center represents mean and SD *** unpaired two-tailed $T$ test; $p = 0.000047$ and 0.00002 $n = 15$ (YQDA mock), 14 (YQDA *ptc-3(RNAi)*), 17 (YDA mock) and 15 (YDA *ptc-3(RNAi)*) animals. Statistical source data are available as source data.

membrane for both probes in *ptc-3(RNAi)* animals compared to mock-treated controls. (Fig. 2D, E). Thus, the strongest cholesterol accumulation is observed in the apical membrane in the intestine of *ptc-3(RNAi)* animals. We envisage that cholesterol levels are also increased, albeit to a lesser extent, in intracellular membranes.

**Low dietary cholesterol rescues *ptc-3(RNAi)* phenotypes**. It is plausible that the cholesterol accumulation is the cause for the observed phenotypes in *ptc-3(RNAi)* animals. *C. elegans* is unable to synthesize cholesterol and must ingest it through the diet[35]. In the lab, cholesterol is provided in the growth medium. Strikingly, when we omitted cholesterol from the growth medium, *ptc-3*

*(RNAi)* worms developed much better, with 89% reaching adulthood (Fig. 3A, B). Moreover, the pale phenotype was strongly reduced, and lipid storage was improved (Fig. 3C, D). Thus, reducing cholesterol accumulation rescued developmental as well as fat content in *ptc-3(RNAi)* worms. Since cholesterol conversion into 7-DHC was also impaired, we asked whether complementing plates with 7-DHC instead of cholesterol would likewise rescue the *ptc-3(RNAi)* phenotypes. However, the addition of 7-DHC resulted in a similar *ptc-3(RNAi)* phenotype than cholesterol depletion (Fig. S2B). We conclude that most of the *ptc-3(RNAi)* phenotypes are linked to the regulation of intracellular cholesterol levels. Moreover, the accumulation of cholesterol, and not the inability of the *ptc-3(RNAi)* animals to

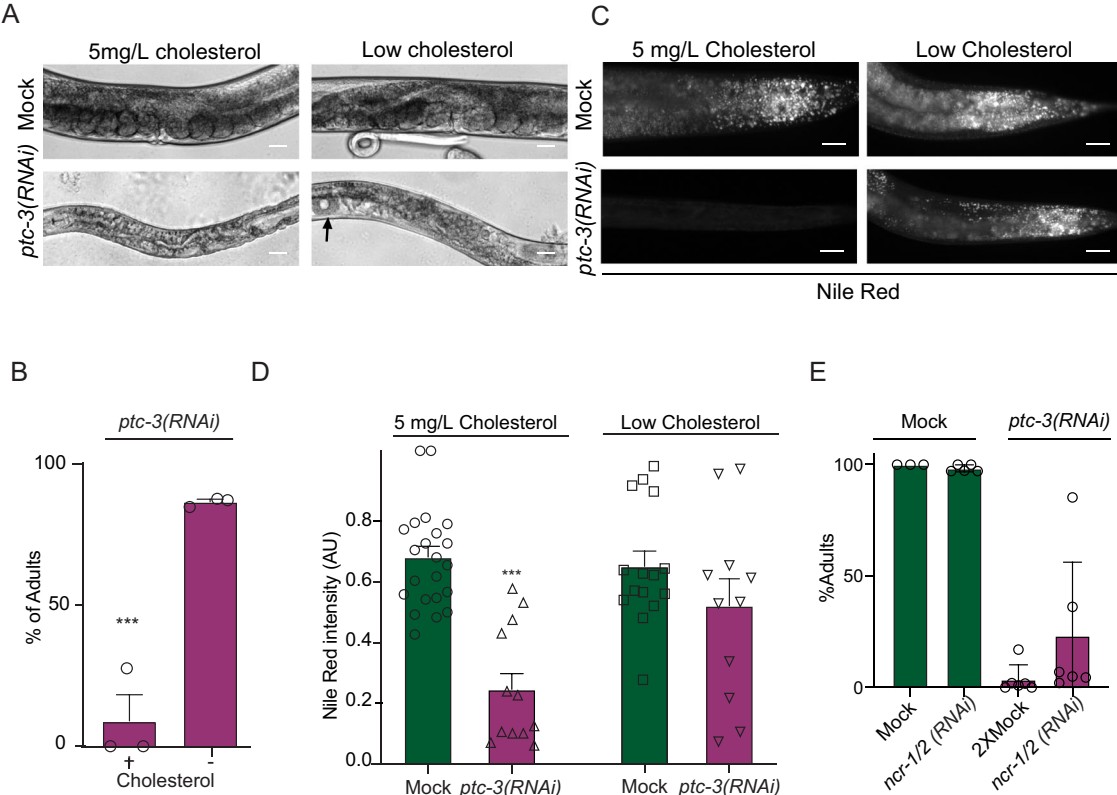

**Fig. 3 Low dietary cholesterol rescues *ptc-3(RNAi)* induced phenotypes.** Worms were fed from L1 larva under standard cholesterol conditions (5 mg/l) or low cholesterol conditions (no added cholesterol) for 3 days. **A** Representative light microscopy images of mock- or *ptc-3(RNAi)*-treated worms on low cholesterol plates Scale bars 20 µm. **B** Quantification of number of *ptc-3(RNAi)*-treated worms that reached adulthood in the absence of cholesterol in the growth medium. Mean and SEM error bars, unpaired one-tailed *T* test ***$p = 0.000565$ $n = 83$ (mock) and 93 (*ptc-3(RNAi)*) animals from four independent experiments. **C** Nile Red staining of lipid droplets in *C. elegans* adult mock- and *ptc-3(RNAi)*-treated after L2 stage on normal or low cholesterol conditions. In low cholesterol conditions, lipid droplet levels are restored in *ptc-3(RNAi)* animals. Scale bars 20 µm. **D** Quantification of Nile Red staining of data shown in (**C**) Mean and SEM error bars. ***$p = 0.000003$ $n = 22$ (mock) $n = 12$ (*ptc-3(RNAi)*) $n = 15$ (low cholesterol mock) and 11 (low cholesterol *ptc-3(RNAi)*) animals. One-way ANOVA including data on Fig. 1C. The normal cholesterol data are the same as depicted in Fig. 1E. **E** Quantification of number of *ptc-3(RNAi)*-treated worms that reached adulthood in combination with *ncr-1/2(RNAi)*. Mean and SEM error bars, $n = 121$ (Mock), 493 (*ncr-1/2 (RNAi)*), 408 (2X Mock), and 295 (*ncr-1/2(RNAi); ptc-3(RNAi)*) animals from three independent experiments. Statistical source data are available as source data.

process cholesterol efficiently, appears to be detrimental for the organism. In mammals, NPC1 mediates cholesterol export from the lysosome[36,37]. The *C. elegans* homologs of NPC1 are NCR-1 and NCR-2. We wanted to test whether blocking the export of cholesterol from the lysosome would alleviate the *ptc-3(RNAi)* phenotype and performed a triple knockdown of NCR-1, NCR-2, and PTC-3. We observed a trend of improvement of the developmental defects, with a relatively big spread, which was therefore not statistically significant (Fig. 3E). Still, our data potentially indicate that reducing cellular available cholesterol by trapping it in lysosomes might be beneficial under these conditions.

**Lipid droplet biogenesis and ER morphology are impaired upon *ptc-3(RNAi)*.** Cellular fat is mostly stored in lipid droplets, which originate from the ER. In *ptc-3(RNAi)* animals, we observed defects in fat content dependent on the intracellular cholesterol levels. Thus, we investigated whether the ER was affected by loss of PTC-3 function using intestinally expressed TRAM-GFP. We observed morphological alterations in the ER in *ptc-3(RNAi)* animals, which were, however, hard to interpret (Fig. S3D). To gain a better understanding of the phenotype, we performed electron microscopy (EM). Not unexpectedly, given the fat content defect, lipid droplets were essentially absent in *ptc-*

*3(RNAi)* intestines (Fig. 4A). Even more strikingly, the ER had lost most of its reticulate structures and formed long lines. Such long lines in 2D are indicative of ER sheets in 3D[38]. We used focused ion beam scanning electron microscopy (FIB-SEM) and machine learning algorithms to obtain information on the ER structure in 3D. Indeed, the reticulate, tubular structure of the ER was dramatically reduced in *ptc-3(RNAi)* when compared to mock; instead enormous ER-sheets and clusters were formed (Fig. 4B, Fig. S4, Movie S1, and S2). Taken together, our data so far suggest that the cholesterol accumulation, due to the absence of PTC-3, impairs ER structure and thereby lipid droplet formation. If the cellular cholesterol levels were indeed the critical factor, then reducing dietary cholesterol in *ptc-3(RNAi)* animals should alleviate the ER phenotype. Indeed, *ptc-3(RNAi)* animals raised on low cholesterol diet displayed reticulated ER and lipid droplets (Fig. 4A). Thus, cellular cholesterol levels strongly influence ER morphology and function. At this point, we were unable to determine whether this effect is direct or indirect. Even though, most of the cholesterol accumulated in the apical plasma membrane in *ptc-3(RNAi)* animals, we cannot exclude, that there is also an accumulation of cholesterol in the ER. Unfortunately, filipin bleaches very fast, and the D4-mCherry sensors are present throughout the cell, so that we are only able to detect very strong local accumulations. Still, the inability of the ER membrane to

A

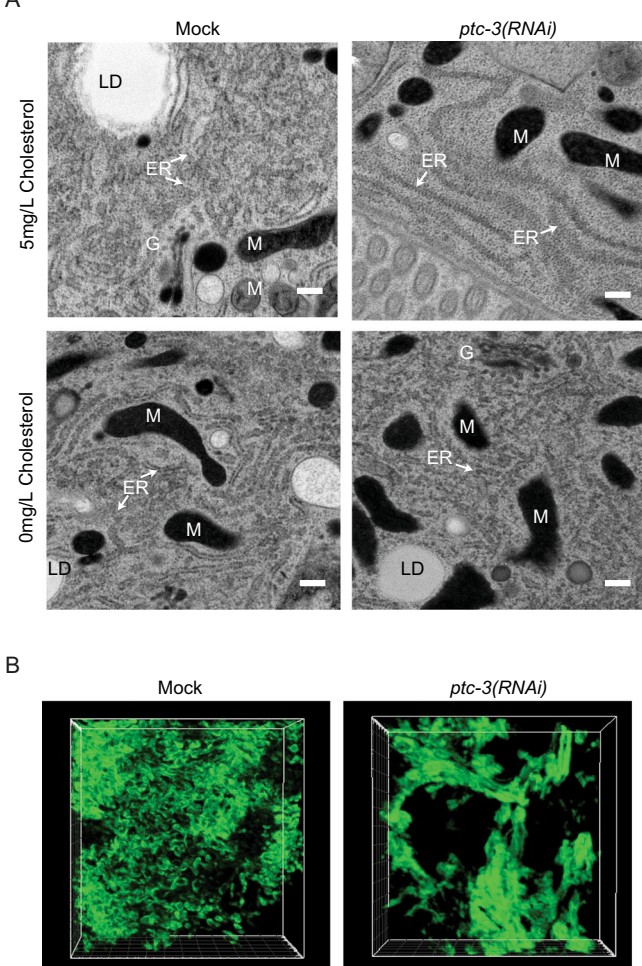

B

**Fig. 4 *ptc-3(RNAi)* reduces LD in the gut and induces changes in the ER structure. A** Transmission electron microscopy (TEM) of adult mock and *ptc-3(RNAi)* treated animals after L2 stage reveal a reduction of reticulate ER structures and LD in *ptc-3(RNAi)* animals. This phenotype is rescued by the omission of cholesterol in the medium. ER endoplasmic reticulum, LD lipid droplet, M mitochondria, G Golgi. Scale bars 200 nm. *n* = 4 (mock) and 6 (*ptc-3(RNAi)*)) animals. **B** Reconstitution of ER membranes from FIB-SEM images of mock and *ptc-3(RNAi)* treated animals using machine learning. *ptc-3(RNAi)* induces sheet-like ER structures.

form lipid droplets and the sheet structure might be linked to the increased membrane bending rigidity.

**Lipid droplet biogenesis defects are not due to increased autophagy or lipolysis or a reduction in Lipid droplet components.** There are other possibilities to explain our findings. For example, lipid uptake could be impaired. First, we did not detect any defects in microvilli organization, apical and basal membrane organization by EM in *ptc-3* (RNAi) animals (Fig. S3A). Moreover, lipid and/or nutrient uptake defects would trigger autophagy. However, we did not observe any increase of autophagy by EM nor did we detect any changes with the *C. elegans* LC3 LGG-1::GFP in the intestine of *ptc-3(RNAi)* worms compared to mock-treated animals (Fig. S3B), making it unlikely that the reduction of lipid droplets is due to impaired lipid uptake. Next, we examined the level of the triglyceride lipase ATGL-1 in *ptc-3* (RNAi) animals. Again, we failed to detect a difference between mock and RNAi-treated animals (Fig. S3B). Finally, we checked the levels of DHS-3, a component of lipid droplets. We did not

detect any significant difference in the DHS-3 protein levels between mock and RNAi-treated animals (Fig. S3B, C). Taken together, our results suggest that the reduction of lipid droplets in *ptc-3(RNAi)* worms is not due to increased autophagy or lipolysis, or the reduction of bona fide lipid droplet components. Moreover, they indirectly support our hypothesis that the ER membrane in *ptc-3(RNAi)* animals might be too stiff to form lipid droplets.

**Fatty acid acyl chain length and desaturation is reduced in *ptc-3(RNAi)* animals.** To test this hypothesis, we first performed a simple experiment, in which we modulated the growth temperature. Membrane fluidity increases as a function of temperature, while membrane bending rigidity decreases. Consistent with our hypothesis, the development and viability of *ptc-3(RNAi)* animals were improved at an elevated temperature (Fig. 5A, B).

Another factor, which determines the stiffness or fluidity of membranes is the saturation of the acyl chains of lipids. Saturated acyl chains are considered to be relatively straight, allowing a high packing rate of lipids accompanied with the generation of an ordered phase and a reduction in fluidity. In contrast, desaturated fatty acids correlate with less dense packing, higher membrane fluidity and lower bending rigidity. Therefore, we performed lipidomics and determined the level of phospholipid acyl chain saturation upon *ptc-3(RNAi)* (Fig. S5A–E). We did not observe any major difference in the headgroup composition of the most important lipid species, except for an increase in PE at the expense of PS and PC (Fig. 5C). Changes in the PC/PE ratio are known to be associated with ER stress[39]. To increase the PC production in order to balance the PC/PE ratio, we fed *ptc-3 (RNAi)* treated animals with 10 mM choline (Fig. S5F)[40]. Choline supplementation had no positive effect on the worm development, suggesting that the changes in PC/PE ratio are not the major cause of the developmental arrest observed in *ptc-3(RNAi)* treated animals. In contrast, we detected a reduction in polyunsaturated fatty acids (PUFAs) in *ptc-3(RNAi)* worms as there was a marked decrease in acyl chain length and desaturation (Fig. 5D). This reduction in PUFAs is not due to a general reduction in lipids upon *ptc-3(RNAi)* compared to mock treatment, but rather reflects a shift from PUFAs to more saturated, shorter FAs. This shift towards more saturated FAs supports our hypothesis that the cholesterol accumulation contributes, directly or indirectly, to the morphological changes of the ER membrane.

**NHR-49 and FAT-7 overexpression rescue *ptc-3(RNAi)* animals.** The reduction in PUFAs could potentially be due to inhibition or lower expression of fatty acid desaturases and elongases. A potential candidate to check this hypothesis is the desaturase FAT-7, which appeared to be down-regulated during heat adaptation to counteract the increase in membrane fluidity at high temperature[41]. Overexpression of FAT-7 in the intestine resulted in better survival of *ptc-3(RNAi)* animals (Fig. 6A, B). The rescued animals were darker than their counterparts (Fig. 6A), suggesting that they were able to store fat. FAT-7 expression is regulated by the PPARα homolog NHR-49[42,43]. Similar to what we had observed for FAT-7 overexpression, increasing intestinal NHR-49 levels improved survival of *ptc-3(RNAi)* animals (Fig. 6A, B). Rescue of survival due to NHR-49 overexpression was accompanied by restoration of fat content (Fig. 6A–D), suggesting that NHR-49 is a major downstream effector of PTC-3. NHR-49 partners with NHR-80, a homolog of mammalian HNF4, to regulate fatty acid desaturation[42]. However, overexpression of NHR-80 did not rescue the *ptc-3(RNAi)* phenotype (Fig. 6A, B).

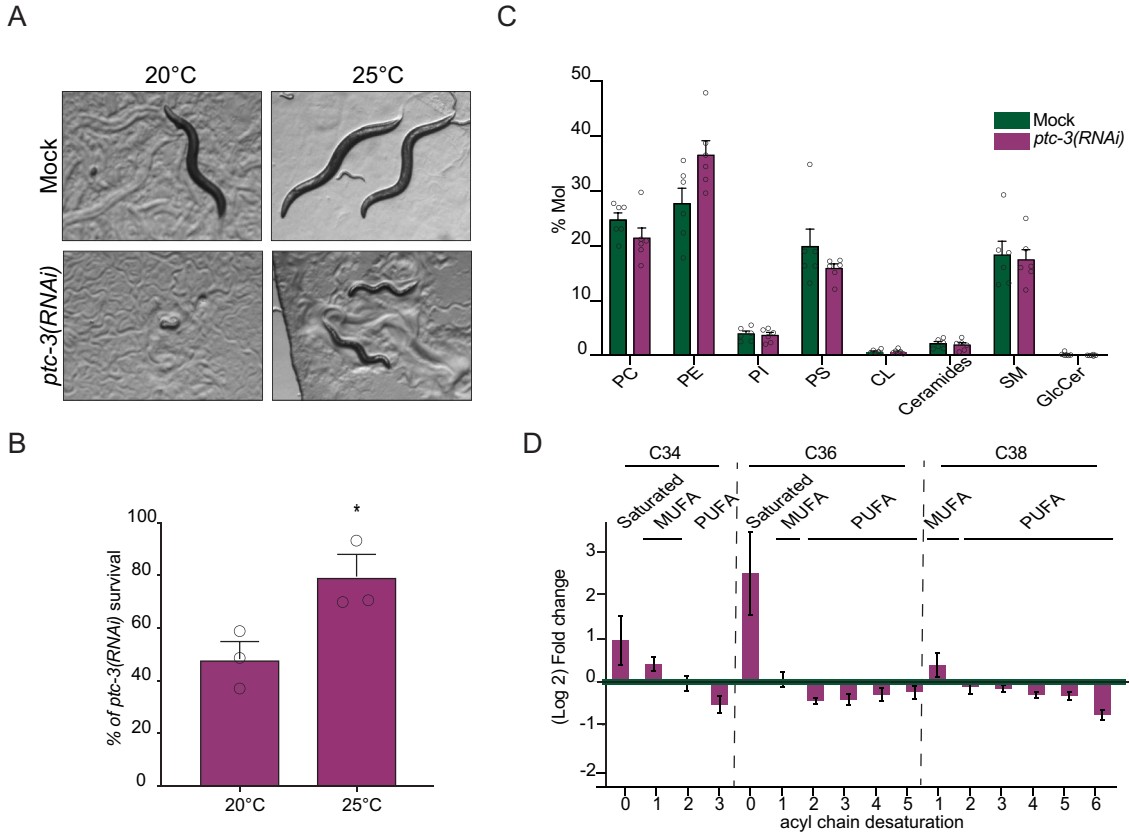

**Fig. 5 ptc-3(RNAi) decreases phospholipid FA saturation and elongation. A** Increasing the growth temperature from 20 °C to 25 °C improves the development of *ptc-3(RNAi)* animals. Representative bright-field pictures of worms on growth plates. **B** Quantification of *ptc-3(RNAi)* survivors at both temperatures. Mean and SEM error bars, unpaired one-tailed *T* test, = 197 (20 °C) and 246 (25 °C) animals from three independent experiments *$p$ = 0.02. **C** Lipidomics on mock or RNAi-treated worms. *ptc-3(RNAi)* worms showed minor differences in lipid head group distribution. Mean and SEM error bars, $n = 6$. **D** Lipidomics revealed differences in lipid acyl chain composition upon *ptc-3(RNAi)*. There is a shift from PUFAs to saturated FA and MUFAs. Mean and SEM error bars, each point represents the average of six experiments per lipid type. Statistical source data are available as source data.

Our data are consistent with the notion that NHR-49 and FAT-7 are modulators of membrane bending rigidity.

**Loss of NHR-181 rescues the high cholesterol-induced phenotypes in *ptc-3(RNAi)* animals.** Nuclear hormone receptors often act context-dependent. Therefore, we wondered, whether other NHRs or the loss thereof may contribute to the *ptc-3(RNAi)* phenotype. NHR-8, the *C. elegans* ortholog of vertebrate liver X and vitamin D receptors, was also shown to influence cholesterol levels and fat content[44,45]. Since *nhr-8(RNAi)* animals contained more fat[45], we speculated whether loss of NHR-8 could rescue the *ptc-3 (RNAi)* phenotype. However, we could not detect any rescue (Fig. S5G). This result may not have been so unexpected since *nhr-8* mutants contain less unsaturated fatty acids[45]. Overexpression of NHR-8 still did not alleviate *ptc-3(RNAi)* defects (Fig. S5H), indicating that NHR-8 and PTC-3 act independently.

*C. elegans* expresses 278 nuclear hormone receptors. To identify possible NHRs important in a PTC-3-dependent pathway, we turned to genome-wide expression data during development. *C. elegans* goes through 4 larval stages before reaching adulthood (Fig. 6E). Each transition from one larval stage to the next is accompanied by the synthesis of a new, larger cuticula, in a process referred to as molting. Genome-wide RNAseq and Riboseq throughout *C. elegans* development revealed an oscillatory behavior of gene expression for many genes[46] (Fig. 6E). Given the general role of PTC-3 in development and the observed cuticle defects upon *ptc-3(RNAi)*, it was not surprising to find that PTC-3 expression also oscillated (Fig. 6F).

However, NHR-49 expression remained constant during development (Fig. 6F). We then asked, which other NHRs would oscillate in a similar manner as PTC-3. Three NHRs emerged as possible candidates: NHR-41, NHR-168, and NHR-181 (Fig. 6G). We hypothesized that the expression levels of the NHRs should be responsive to cholesterol levels. Of the three, only expression levels of the HNF4 homolog NHR-181 were upregulated in high cholesterol, i.e., *ptc-3(RNAi)*, and reduced under low cholesterol conditions (Fig. 6H). More importantly, knockdown of *NHR-181* rescued the *ptc-3(RNAi)* induced lethality to a similar extent than overexpression of NHR-49, irrespective of the cholesterol present in the medium (Fig. 6B, I). Moreover, fat content was restored to a similar extent (Fig. 6J, K, compare J and C). Taken together, our data imply that NHR-49 positively and NHR-181 negatively regulate membrane bending properties and fat content in response to high cholesterol levels.

## Discussion

We explored the role of the *C. elegans* PTCH homolog, PTC-3, in the absence of the classical functional Hh signaling pathway. The function of PTCH proteins is conserved from *C. elegans* to man because similar to what has been proposed for mammalian PTCH[1,6,47], PTC-3 is a cholesterol transporter, which exports cholesterol out of the cell. PTC-3 appears to be the major cholesterol transporter in the apical plasma membrane in the *C. elegans* intestine, since knock-down of PTC-3 resulted in strong intracellular cholesterol accumulation, most notably in the apical plasma membrane.

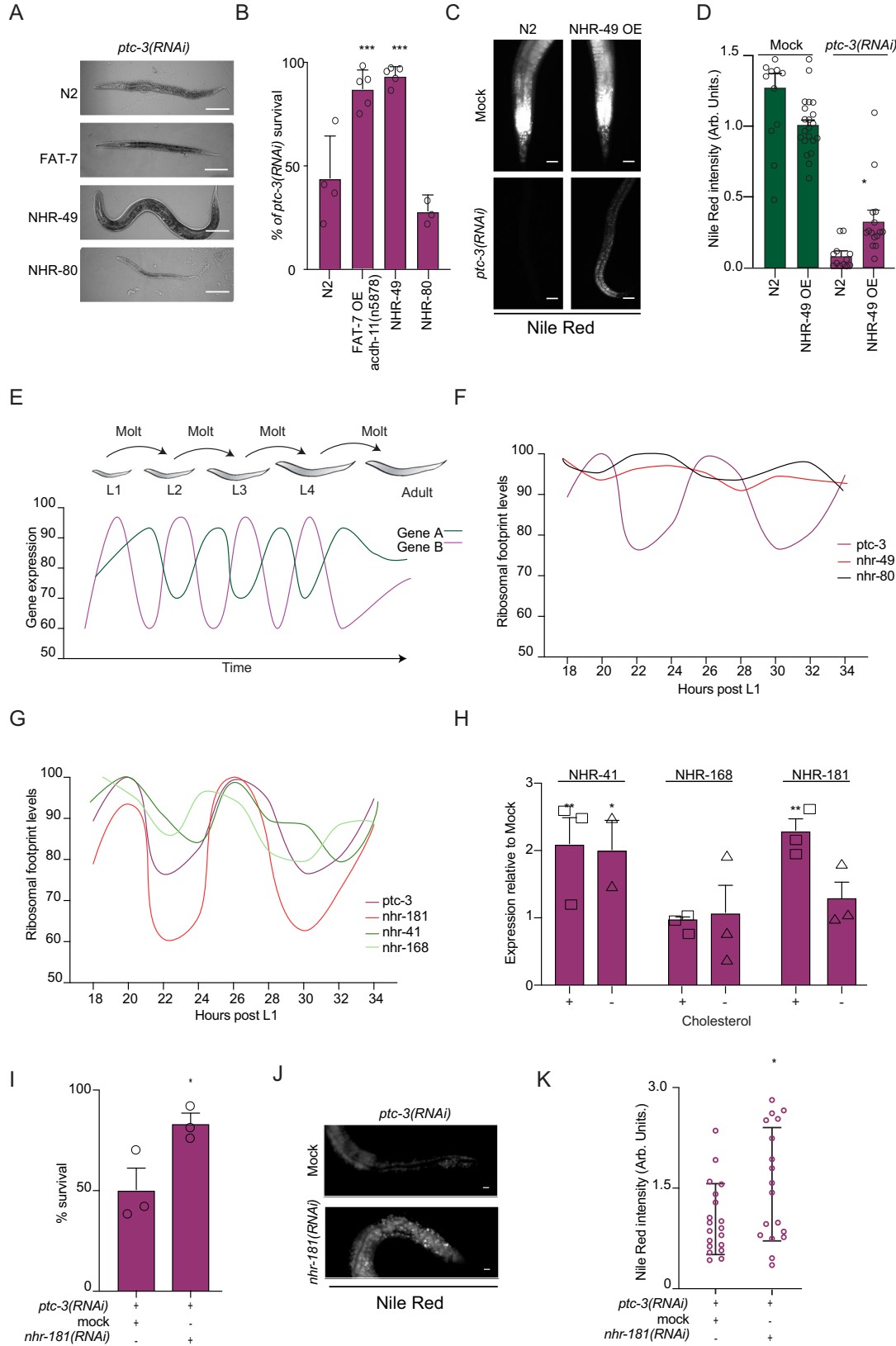

As a consequence of the cholesterol accumulation, the balance of tubular and sheet-like ER was strongly skewed towards sheet structures (Fig. 7). Moreover, lipid droplet synthesis, which originates at the ER was greatly reduced. We propose that the ER membranes in *ptc-3(RNAi)* animals have an increased bending rigidity, which does not allow bulging out of lipid droplets. Whether triglycerides still accumulate within the lipid bilayer remains unclear because their detection in the ER membrane was not possible. However, we observed an increase in saturated and mono-unsaturated fatty acid acyl chains (MUFAs) at the expense

**Fig. 6 PTC-3 influences NHR function in a cholesterol-dependent manner. A** Overexpression of NHR-49 or FAT-7 partially rescues *ptc-3(RNAi)* defects. Representative DIC images of worms. Scale bars 100 μm. **B** Quantification of survival rate upon overexpression of FAT-7, NHR-49 or NHR-80 in *ptc-3 (RNAi)* animals. Mean and SD error bars. FAT-7 ***p = 0.0004, NHR-49 ***p = 0.0001. (**C**) Over expression of NHR-49 partially restores fat accumulation in *ptc-3(RNAi)* animals. Nile Red staining. Scale bars 20 μm. (**D**) Quantification of data shown in (**C**) *p = 0.0496 n = 11 (Mock N2), 21 (Mock; NHR-49 OE), 12 (*ptc-3(RNAi)* N2) and 15 (*ptc-3(RNAi)* NHR-49 OE) animals from 3 independent experiments. Mean and SEM error bars. **E** Schematic representation of ribosomal footprints of mRNA during *C. elegans* larval development. Oscillatory changes in mRNA levels during developmental time. The timing, amplitude, and whether a gene is oscillating is gene specific. **F** PTC-3, but not NHR-49 or NHR-80, expression oscillates during development. Data plotted from[46]. **G** Ribosomal footprint oscillations of NHR-181, NHR-168, and NHR-41 are similar to PTC-3 (data from ref. [46]). (**H**) NHR-181 expression is modulated dependent on cholesterol levels. qRT-PCR analysis of NHR-41, NHR-168, and NHR-181 in the presence or absence of cholesterol in the growth medium. Mean and SEM error bars. NHR-41 **p = 0.004 *p = 0.0144, NHR-181 **p = 0.0011. **I** Genetic interaction between PTC-3 and NHR-181. Knockdown of NHR-181 rescues *ptc-3(RNAi)* lethality. Error bars are SEM *p = 0.0496 n = 244 (Mock; *ptc-3(RNAi)*) and 164 (*nhr-181(RNAi)*; *ptc-3(RNAi)*) animals from three independent experiments. **J** *nhr-181(RNAi)* partially restores fat accumulation in *ptc-3(RNAi)* animals. Nile Red staining. Scale bars 20 μm. **K** Quantification of data shown in (**J**) *p = 0.0306 n = 19 (Mock; *ptc-3(RNAi)*) and 18 (*nhr-181(RNAi);ptc-3(RNAi)*) animals from three independent experiments. Mean and SD error bars. Statistical source data are available as source data.

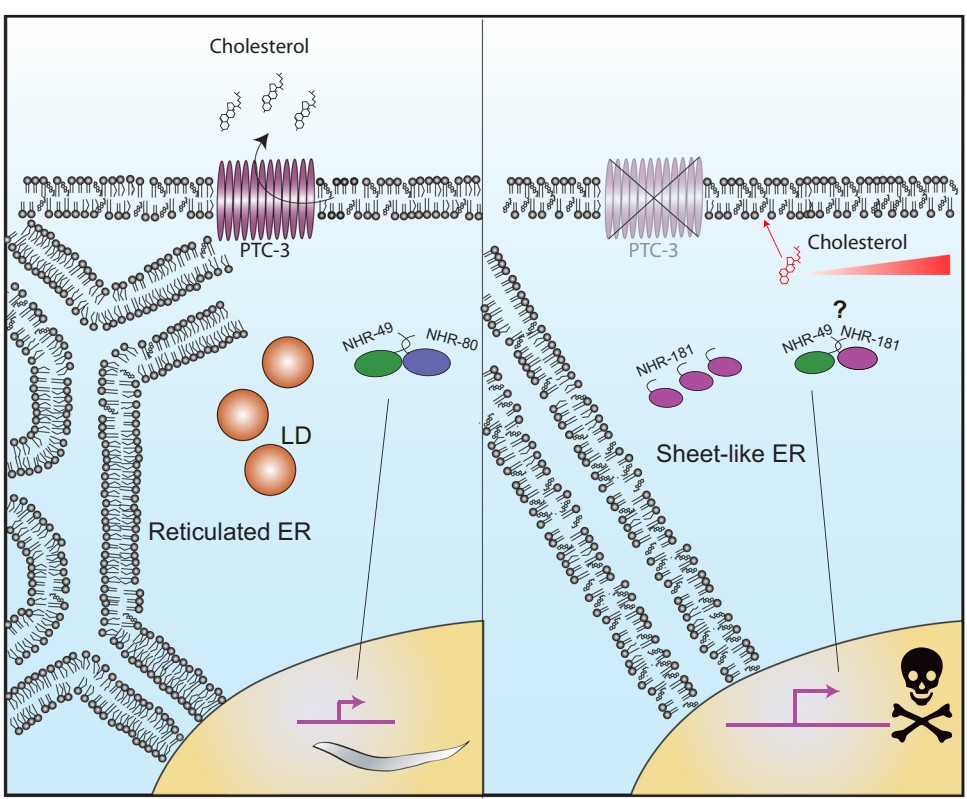

**Fig. 7 Model of how PTC-3 and the loss thereof affects ER structure and LD formation.** PTC-3 controls intracellular cholesterol levels directly by promoting its efflux. In the absence of PTC-3, cells accumulate cholesterol, which in turn directly influences membrane properties. In addition, cholesterol directly or indirectly affects NHRs, which subsequently leads to a reduction of acyl chain length and desaturation. This second effect enhances the changes in membrane properties and leads to changes in ER morphology and LD formation. ER endoplasmic reticulum, LD lipid droplet.

of PUFAs upon knockdown of PTC-3. This imbalance towards shorter and saturated acyl chains should also increase lipid packing and increase membrane bending rigidity. This effect would then be intensified by the accumulation of cholesterol in intracellular membranes, which would furthermore promote membrane stiffness.

While the increase of membrane bending rigidity is probably sufficient to inhibit lipid droplet formation, vesicle formation at the ER and the Golgi apparatus may not be as strongly affected by cholesterol accumulation and the increase in MUFAs because we still observed stacked Golgi apparatus by EM (Fig S6). The difference between lipid droplet formation and vesicle budding is that the COPII coat can bend the entire lipid bilayer[48], while the triglycerides must deform the membrane from within and push

the lipid bilayer apart, a process, which might be less energetically favorable. The COPII coat can thus probably exert the force necessary to bend the ER membranes in the mutant.

However, the cholesterol accumulation may not only have a structural role in stiffening of membranes, together with MUFAs and saturated fatty acid acyl chains. It was shown previously that cholesterol and downstream metabolites can act as hormones in *C. elegans*[49–51]. In support of this notion, we observed cuticular defects with intestine-specific knockdown on PTC-3, indicating that there are cell non-autonomous effects of *ptc-3(RNAi)*. Given that PTC-3 is a cholesterol transporter, we speculate that the increased cholesterol levels in the intestine and lack of organismal distribution of cholesterol to hormone-forming tissues are the causative of the cuticle defects. Moreover, cholesterol may also

change the transcriptional program and reduce the expression of genes required for FA desaturation and elongation. In fact, overexpression of the PPARα homolog NHR-49 rescued the *ptc-3* (*RNAi*)-induced fat loss and developmental arrest phenotype. In mammalian cells, PPARα requires PUFAs for activation[41,52]. We hypothesize that reduced PUFA levels downregulate NHR-49 activity, which could be compensated by the overexpression of NHR-49. Alternatively, but not mutually exclusive, the interaction between NHR-49 and NHR-80, which are jointly controlling FA elongation and desaturation[42], would be disrupted by high cholesterol levels and NHR-49 would instead team up with NHR-181. This complex could then negatively regulate FAT and ELO gene expression when cholesterol levels increase in the cell. A circumstantial argument that puts weight on this latter possibility is that both NHR-80 and NHR-181 are homologs of mammalian HNF4 proteins. Thus, it is tempting to speculate that the exchange of one HNF4 like molecule for another would shift the activity of NHR-49 from promoting elongation and desaturation to repressing these processes. Interestingly, a C-terminal truncation in Ptch1 in adult mice led to a reduction of white fat tissue and PPARγ levels, suggesting that the SMO-independent pathway we uncovered might be conserved in mammals[53].

NHR-49 and NHR-181 appear to be specific downstream effectors of PTC-3 activity levels, as downregulating NHR-8 did not improve PTC-3-dependent phenotypes. Likewise, SREB, which is a major responder to alteration in cellular cholesterol levels and which has been shown to regulate the expression of fatty acid elongases and desaturases in mammalian cells[54]. However, knockdown of PTC-3 did not affect the nuclear localization of the *C. elegans* SREB homolog SBP-1 (Fig. S5I). Yet, NHR-49 activity clearly is affected by increased cholesterol levels since overexpression of its target and activator FAT-7[41] partially rescued *ptc-3*(*RNAi*) phenotypes. How the entire lipidome is affected under these conditions remains to be determined.

We used *C. elegans* to reveal potential ancestral functions of PTCH family proteins because it lacks smoothened and other canonical Hh signaling pathway components, which have been lost during evolution[22,23]. In fact, it has been proposed that PTCH and related proteins such as NPC1 and dispatched evolved separately from smoothened[55]. PTCH belongs to the family of RND transporters, which are already present in bacteria. For most bacterial RNDs the substrates are unknown. However, the family most related to PTCH transports hopanoids, which are structural and functional analogs of sterols[55]. Next to PTC-3, *C. elegans* encodes another PTCH protein PTC-1 and 23 PTCH-related proteins (PTRs), presumably all RND transporters. It is tempting to speculate that these PTCs and PTRs are transporting small molecules, presumably sterols, and thereby contributing to cellular homeostasis and potential intra- and intercellular communication.

Our data strongly implicate cellular cholesterol levels, membrane composition, and nuclear hormone receptors such as PPARα and HNF4 in non-canonical Hh signaling pathways. They also provide a framework on how to distinguish between SMO-dependent and -independent functions in mammals. Our results might be particularly important for the understanding of diseases such as multiple myeloma in which canonical and non-canonical Hh signaling have been implicated[56,57].

## Methods

**General methods and strains.** *C. elegans* was cultured and maintained as described previously[58] at 20 °C unless it was specified different. RNAi was carried out using sequenced and confirmed clones from the Ahringer library, as mock nontargeting dsRNA from the Ahringer library clone Y95B8A_84.g was used[59]. For low cholesterol conditions cholesterol was omitted and agar replaced by agarose. RNAi feeding experiments were performed for 3 days starting from L1 larvae. When adult *ptc-3*(*RNAi*) were needed worms were grown with RNAi mock bacteria until L2 stage and then transferred to *ptc-3*(*RNAi*) plates for 2 days. For developmental and survival assays eggs from 1-day adult worms were hatched in M9 buffer (3 g KH$_2$PO$_4$, 6 g Na$_2$HPO$_4$, 5 g NaCl, 1 ml 1 M MgSO$_4$, H$_2$O to 1 l; sterilized by autoclaving) overnight without bleaching. L1s were transferred to RNAi plates and grown at 20 °C. Survival and developmental stage were assessed after 72 h. For double RNAi experiments *ptc-3*(*RNAi*) was diluted 1:1 with the second RNAi or mock expressing bacteria. *nhr-49(nr2041)* [*ges-1p::3xHA::nhr-49(cDNA)::unc-54 3'UTR + myo-3p::mCherry::unc-54 3'UTR*] and for gut-specific RNAi *kbIs7* [*nhx-2p::rde-1 + rol-6(su1006)*] was used, *sbp-1(ep79)* [*sbp-1::GFP::SBP-1; rol-6(su1006)*], were obtained from the Caenorhabditis Genetics Center (CGC). *nhr-8(hd117)* mutant and nhr-8::GFP over expressing strains were described previously[45]. For the cholesterol sensor strains generation the PFO-derived D4 domain mutants YDA (D434W Y415A A463W) and YQDA (D434W Y415A A463W Q433W) fused to a mCherry N-terminal tag were cloned using the NEBuilder HiFi DNA Assembly Cloning Kit (NEB #E5520) and introduced into pBlueScriptII with a VHA-6 promoter and tub terminator using the primers pvha6_fwd and pvha6_rev for the promoter amplification, for the sensor amplification D4H_fwd and D4H_rev and tubter_fwd and tubter_rev for tubulin terminator were used (Table S1). The plasmid was microinjected at a concentration of 50 ng/μl into both arms of the syncytial gonads of N2 worms. SUR-5::GFP at 10 ng/μl concentration was co-injected as transformation marker and 40 ng/μl of lambda DNA as carrier was used. Animals containing the cholesterol sensors were grown at 25 °C and fed with OP50 RNAi-competent bacteria[60]. The *ptc-3::gfp* reporter pCH115.1 was constructed by inserting a gfp cassette into the same site within the *ptc-3* locus as described in ref. [18] in the fosmid WRM064cC06, following the recombineering protocol described in ref. [61]. The gfp cassette was PCR-amplified by using pBALU1 vector and primers CH428 and CH429 (Table S1), and the *galK* module was excised after its insertion to the fosmid. As co-injection marker pMF435: pgp-1:: mCherry::unc-54 3'UTR was used[62].

**Microscopy.** Live worms were immobilized with 50 mM levamisole in M9 and mounted on a slide with 2% agarose. The worms were imaged with a Zeiss Axioplan 2 microscope equipped with a Zeiss Axio Cam MRm camera (Carl Zeiss, Aalen Oberkochen, Germany), and the objectives EC Plan-Neofluar 10x/0.3, EC Plan-Neofluar 20x/0.50, EC Plan-Neofluar 40/1.30. All images were adjusted to the same parameters with OMERO.web 5.3.4-ice36-b69. Images of D4H cholesterol sensors, Filipin III staining and TRAM::GFP were obtained on a Zeiss LSM 880 microscope with Airyscan with Plan-Apochromat 63x/1.4 Oil DIC M2. The fast mode was used, and images were processed using the Zen Black software.

**Coherent anti-stokes Raman spectroscopy.** Worms were mounted on a slide with 2% agarose with 20 mM levamisol. A Leica TCS SP8 system with a CARS laser picoEmerald was used. The lasers were beamed to 816.4 nm while keeping the Stokes beam constant at 1,064.6 nm. The scan speed was set to 400 Hz. A z-stack per worm was imaged along the intestine and 19 animals from three experiments were collected per condition. The number of lipid droplets in each stack was assessed with the Fiji plug-in Lipid Droplet Counter. The data were analyzed with a one-tail ANOVA followed by Dunnett's multiple comparisons test in Prism 7.

**TEM and FIB SEM.** For transmission electron microscopy (TEM) and FIB-SEM, worms were frozen as follows. *C. elegans* animals were picked with a worm pick from an agar plate and transferred to a droplet of M9 medium on a 100 μm cavity of a 3 mm aluminum specimen carrier (Engineering office M. Wohlwend GmbH, Sennwald, Switzerland). 5–10 worms were added to the droplet and the excess M9 medium was sucked off with dental filter tips. A flat aluminum specimen carrier was dipped in 1-hexadecene and added on top. Immediately, the specimen carrier sandwich was transferred to the middle plate of an HPM 100 high-pressure freezer (Leica Microsystems, Vienna, Austria) and frozen immediately without using ethanol as synchronizing medium.

Freeze-substitution was carried out in water-free acetone containing 1% OsO$_4$ for 8 h at −90 °C, 7 h at −60 °C, 5 h at −30 °C, 1 h at 0 °C, with transition gradients of 30 °C/h, followed by 30 min incubation at RT. Samples were rinsed twice with acetone water-free, block-stained with 1% uranyl acetate in acetone (stock solution: 20% in MeOH) for 1 h at 4 °C, rinsed twice with water-free acetone, and embedded in Epon/Araldite (Merck, Darmstadt, Germany): 66% in acetone overnight, 100% for 1 h at RT and polymerized at 60 °C for 20 h. Ultrathin sections (50 nm) were post-stained with Reynolds lead citrate and imaged in a Talos 120 transmission electron microscope at 120 kV acceleration voltage equipped with a bottom-mounted Ceta camera using the Maps software (Thermo Fisher Scientific, Eindhoven, The Netherlands).

For Focused ion beam scanning electron tomography (FIB-SET), a trimmed Epon/Araldite block containing a single *C. elegans* was mounted on a regular SEM stub using conductive carbon and coated with 10 nm of carbon by electron beam evaporation to render the sample conductive. Ion milling and image acquisition were performed simultaneously in an Auriga 40 Crossbeam system (Zeiss, Oberkochen, Germany) using the FIBICS Nanopatterning engine (Fibics Inc., Ottawa, Canada). A large trench was milled at a current of 16 nA and 30 kV, followed by fine milling at 240 pA and 30 kV during image acquisition with an advance of 5 nm per image. Prior to starting the fine milling and imaging, a

protective platinum layer of ~300 nm was applied on top of the surface of the area of interest using the single gas injection system at the FIB-SEM. SEM images were acquired at 1.9 kV (30 μm aperture) using an in-lens energy selective backscattered electron detector (ESB) with a grid voltage of 550 V, and a dwell time of 1 μs and a line averaging of 130 lines. The pixel size was set to 5 nm and tilt-corrected to obtain isotropic voxels. The final image stack was registered and cropped to the area of interest using the Fiji image-processing package [https://imagej.net/TrakEM2]. FIB-SEM images were processed with iLastik[63] and pixel classification was done. The classifier was trained to separate different object classes, ER, cytoplasm, and other organelles. The training was done individually for each dataset. A 3D reconstruction was later handled with IMARIS 9.2.

**Lipidomic analysis**. Worms were cultured in liquid media as described previously[64]. Feeding bacteria were prepared by growing RNAi bacteria to an $OD_{600}$ of 0.6 in LB-Amp medium and then inducing dsRNA expression with 1 mM IPTG for 24 h. Bacteria were harvested and resuspended to $OD_{600}$ 400. Synchronized populations of worms were grown from L1 larvae to L2 stage in mock bacteria and then transferred into RNAi bacteria until they reached early adulthood. Young adults were collected and washed once in ddH₂O. 8000 young adults were used for glycerophospholipid and sphingolipid analysis while sterol analysis was done from 40,000 young adults. Pellets were frozen and stored at −80 °C until extraction. Lysis was performed on a Cryolysis machine (Precellys 24, lysis & homogenization machine (Bertin Technologies)) at 4 °C using 100 μl 1.4 mm zirconium oxide beads in 800 μl MS-H₂O with three cycles of 45 s bursts at 6200 rpm followed by 45 s interruptions. Lysates were eluted into glass tubes with lipid standards (glycerophospholipid and sphingolipid standards: di-lauryl phosphatidylcholine, di-lauryl phosphatidylethanolamine, di-lauryl phosphatidylinositol, di-lauryl phosphatidylserine, tetra-lauryl cardiolipin, C17 ceramide, C12 sphingomyelin, C8 glucosylceramide, all from Avanti Polar Lipids; sterol standard: ergosterol from Fluka) and beads were washed and eluted again with 200 μl MS-H₂O. Lipids were extracted with chloroform and methanol according to a published protocol[65], with minor modifications. Briefly, 3.6 ml organic solvent (CHCl₃/MeOH = 1:2, v:v) were added to the 1 ml aqueous lysate, mixed and centrifuged to clear extract from worm debris. Extracts were transferred to new glass tubes and phase separation was induced by the addition of 0.5 mL MS-H₂O and 0.5 ml CHCl₃. Samples were centrifuged, and the organic phase was collected. For sterol analysis total lipid extract was dried directly in a centrivap. In order to concentrate and to separate sterols from other lipids solid-phase extraction on a Chromabond® SiOH column (Macherey-Nagel, Germany) was performed. Columns were washed two times with 1 ml CHCl₃. Total lipid extract from 40,000 worms was resuspended in 250 μl CHCl₃ by vortexing and sonication. The extract was then applied to the column and eluted with two times 650 μl CHCl₃. The flow-through and CHCl₃ elutions were combined, dried, and used for sterol analysis by GC-MS (neutral lipid fraction). In the case of glycerophospholipid and sphingolipid analysis, total lipid extract from 8,000 animals was split in two and dried. One aliquot (total lipid fraction) was used without further treatments for glycerophospholipid analysis and inorganic phosphate determination while the other underwent methylamine treatment and desalting via butanol extraction (sphingolipid fraction)[66]. Briefly, the sphingolipid fraction was deacylated to eliminate phospholipids by methylamine treatment[67]. 0.5 mL monomethylamine reagent (MeOH/H₂O/n-butanol/CH₃NH₂ solution; 4:3:1:5 v/v) was added to the dried lipids, followed by sonication (5 min). Samples were then mixed and incubated for one hour at 53 °C and dried under nitrogen. The monomethylamine treated lipids were desalted by n-butanol extraction. 300 μl H₂O saturated n-butanol was added to the dried lipids. The sample was vortexed, sonicated for 5 min, and 150 μl MS grade water was added. The mixture was vortexed thoroughly and centrifuged at 3200 × g for 10 min. The upper phase was transferred in a 2 ml amber vial. The lower phase was extracted twice more with 300 μl H₂O saturated n-butanol and the upper phases were combined and dried under nitrogen.

Glycerophospholipid and sphingolipid analysis was performed following a worm-adapted version of a previously published method[66,68]. LC-MS or HPLC grade solvents were used and the samples were pipetted in a 96 well plate (final volume = 100 μl). Positive mode solvent: CHCl₃/MeOH/H₂O (2:7:1 v/v) + 5 mM NH₄Ac. Negative mode solvent: CHCl₃/MeOH (1:2 v/v) + 5 mM NH₄Ac. The total lipid and sphingolipid fractions were resuspended in 250 μl CHCl₃/MeOH (1:1 v/v) and sonicated for 5 min. The glycerophospholipids (total lipid fraction) were diluted 1:10 in negative and positive mode solvents and the sphingolipids were diluted 1:5 in positive mode solvent and infused onto the mass spectrometer. Tandem mass spectrometry for the identification and quantification of glycerophospholipid and sphingolipid molecular species was performed using multiple reaction monitoring (MRM) with a TSQ Vantage Triple Stage Quadrupole Mass Spectrometer (Thermo Fisher Scientific, Bremen, Germany) equipped with a robotic nanoflow ion source, Nanomate HD (Advion Biosciences, Ithaca, NY). The collision energy was optimized for each lipid class based on the internal standards, and the following m/z transitions were measured using an m/z window of ±0.5 amu for the precursors: in positive electron spray ionization mode (ESI+) phosphatidylcholines M+H+ (Q1) -> 184.07 (Q3), phosphatidylethanolamine M+H+ (Q1) -> neutral loss of 141.02 (Q3), in negative electron spray ionization mode (ESI-) phosphatidylinositol M-H+(Q1)->241.01 (Q3), phosphatidylserine M-H+ (Q1) -> neutral loss of 87.03 (Q3) and cardiolipin M-2H+/2 (Q1) ->

different fatty acid fragments (Q3). Each biological replicate was read in two technical replicates each comprising three measurements for each transition. Lipid concentrations were calculated relative to the corresponding internal standards and then normalized to the total phosphate content of each total lipid extract. Lipid concentrations of PC, PE, PI, and PS were corrected for Class II isotopic over-laps which are almost entirely due to ¹³C isotopes present in natural lipid species[69]. Data parsing, deisotoping, and analysis were done using a custom pipeline in Python v3.8.5[70]. Correction factors for de-isotoping were derived using theoretical M+2 abundances calculated using the Envipat Web 2.4 tool (https://www.envipat.eawag.ch/) applying a mass resolution of 5000. As MRM analysis was set to only filter for non-heavy fragments in the third quadrupole, these theoretical M+2 abundances were multiplied by a correction factor accounting for the probability at random distribution of two ¹³C isotopes within the remaining heavy fragment generated during the fragmentation in the collision chamber (Q2) but not detected in the Q3. The resulting formula for correction is: $M + 2_{correction} = (M + 2_{theoretical})$ $*((n_{heavy})/(m_{total}))^2$ with $n_{heavy}$ being the number of carbons in the heavy fragment, and $m_{total}$ the number of carbons in the entire lipid molecule. For each lipid species the corrected M + 2 signal was calculated and subtracted from the acquired signal for the lipid species with $m/z + 2$ within a series of lipid species from the same lipid class, beginning with the most desaturated species, stepwise until reaching the fully saturated form. In addition, as different biological replicates showed variations in total lipid amounts within each condition, lipid amounts were normalized to reporting Mol% values. The data were analyzed with a two-way ANOVA followed by Šidák's multiple comparisons test in Prism 9.

Sterol analysis was done as previously described[49]. In short, sterol fractions from 40,000 young adults were resuspended in chloroform/methanol (2:1 v/v) and injected into a VARIAN CP-3800 Gas Chromatograph equipped with a Factor Four Capillary Column VF-5 ms (30 m × 0.25 mm ID DF = 0.25) and analyzed by a Varian 320 MS triple quadrupole with electron energy set to −70 eV at 200 °C and the transfer line at 280 °C. The temperature was held for 4 min at 45 °C, ramped successively to 195 °C (20 °C/min), to 230 °C (4 °C/min), to 325 °C (20 °C/min), to 350 °C (6 °C/min) before cooling back to 45 °C. Free sterols were eluted during the linear gradient from 195 to 230 °C. Cholesterol ($m/z = 386.3$), 7-dehydrocholesterol ($m/z = 384.3$) and 4α-methyl-5α-cholest-8(14)-en-3β-ol/lophenol ($m/z = 400.3$) were identified by comparison of elution profiles and fragmentation patterns with reference to the NIST database and cholesterol as well as 7-dehydrocholesterol standards. Quantities of the sterols were determined by extraction of the 384.3, 386.3, and 400.3 ions and normalization via the internal standard, ergosterol ($m/z = 398.3$).

**Nile Red staining**. Nile Red staining was performed as described previously[71]. Worms were washed with 1 ml M9 into a 1.5 ml siliconized microfuge tube. Worms were allowed to sink by gravity on ice and were washed with M9. Approximately 30 μl of M9 and worms at the bottom of the tube were left. 0.2 ml of 40% isopropanol was added and incubated for 3 min for fixation. The fixative was removed and 150 μl of Nile Red solution (6 μl of Nile Red 0.5 mg/ml in acetone per 1 ml of 40% isopropanol) was added to the worms for 30 min at 20 °C with gentle rocking in the dark. Worms were washed once with 1 ml M9 buffer and mounted on a 2% agarose pad for microscopy. Intensity analysis was performed using Fiji with at least 11 worms per condition from three different experiments. The data were analyzed with a one-way ANOVA followed by Dunnett's multiple comparisons test in Prism 7.

**TopFluor cholesterol staining**. The experiment was performed as described previously as[5]. PTC-3 CDS was amplified with the primers ptc-3TEF_F 5′-actagtggatcccccgggctgcaggATGAAGGTGCATTCGGAACAAC-3′ ptc-3TEF_R 5′-gacggtatcgataagcttgatatcgTTACTTGTGCGCTGGCGATG-3′ from cDNA and cloned into the yeast plasmid p426TEF. A point mutation (D697A) was introduced with the Q5® Site-Directed Mutagenesis Kit (E0554S NEB). Yeasts were cultured to an $OD_{600}$ of 4, washed with cold water, and resuspend to 10 $OD_{600}$ in 50 mM HEPES buffer pH 7.0. Yeasts were incubated protected from light with 5 μM TopFluor® Cholesterol (810255, Avanti Polar Lipids) for 2 h at 20 °C. They were washed once with cold ddH₂O and resuspend with HEPES buffer, after 20 min the yeast was spun down, and the supernatant was measured with filters 485ex 520em on a plate reader (DTX880, Multimode Detector, Beckman Coulter). The efflux was normalized to the initial fluorescence of the yeast cells. For the cell-based assay, HEK293 cells were cultured in DMEM (Sigma) high glucose medium with 10% FCS (Bioconcept), 1% penicillin–streptomycin, 1% sodium pyruvate, and 1% L-glutamine. pcDNA-h-mmPtch1-FL was a gift from Philip Beachy (Addgene plasmid # 120889), as a control pcDNA3.1 plasmid was used and C. elegans PTC-3 cDNA was cloned with NEBuilder HiFi DNA Assembly Cloning Kit (NEB #E5520) into pcDNA3.1 with the primers: VectorFL_fwd, VectorFL_rev, PTC-3_fwd and PTC-3_rev (Table S1). One day prior to transfection, cells were plated with 60–70% of confluence. Cells were transfected using Helix-IN™ transfection reagent (OZ Biosciences) according to the manufacturer's instructions. One μg of DNA was used for a 10 cm dish. After 24–48 h post-transfection, cells were incubated with 2.5 μM TopFluor® Cholesterol (810255, Avanti Polar Lipids) for 2 h at 37 °C. Cells were washed once with PBS and incubated with PBS supplemented with 5 mM Dextrose, 1 mM CaCl₂, 2.7 mM KCl, and 0.5 mM MgCl₂. 150 μL samples were collected at times 0, 5, 10, 45, and 60 min. The samples were spun down and 100 μL

of the supernatant was measured with filters 485ex 520em on a plate reader (DTX880, Multimode Detector, Beckman Coulter). For the analysis in *C. elegans*, worms were washed off a plate with M9 buffer and put on a shaker in M9 buffer with 5 mM TopFluor® Cholesterol for 1 h at 20 °C. Worms were washed with M9 buffer once to remove the excess of Topfluor® Cholesterol and chase in M9 buffer was performed for 1 h before imaging.

**Immunofluorescence and PTC-3 antibody**. Immunofluorescence of *C. elegans* was performed as described previously[72], with slight modifications: Worms were blocked with PTB (1% BSA, 1× PBS, 0.1% Tween20, 0.05% NaN₃,1 mM EDTA) and secondary antibody was diluted in PTB. Peptide antibodies against *C. elegans* PTC-3 were generated in rabbits by Eurogentec using peptides SASHSSDDES-SPAHK and EVRRGPELPKENGLG. Serum was used in a 1:100 dilution and Alexa Fluor 488-goat anti-rabbit IgG (H+L) (Invitrogen; A-11034) 1:5000. Worms were washed 2× in M9 and mounted with fluorescence protecting media (ProLong™ Glass Antifade Mountant Invitrogen P36984). Worms were imaged on a Zeiss LSM 880 microscope as described in the Microscopy section.

**Filipin staining**. Worms were fixed in Glyoxal solution (2.835 ml ddH₂O, 0.789 ml EtOH, 0.313 ml glyoxal (40% stock solution from Sigma-Aldrich, #128465) 0.03 ml glacial acetic acid. pH 4.5) for 30 min on ice, and for another 30 min at RT, followed by 30 min of quenching in 100 mM NH₄Cl at RT and O/N post quenching at 4 °C[73]. Worms were washed 2 × 30 min with M9 and left in 50 µl of M9 in which 50 µl of Filipin III ready-made solution (Sigma-Aldrich, SAE0087) was added for 1 h in the dark at RT. Worms were washed 2× in M9 and mounted with fluorescence protecting media (ProLong™ Glass Antifade Mountant Invitrogen P36984). Worms were imaged on a Zeiss LSM 880 microscope as described in the Microscopy section.

**Western blot**. Worm Lysate from synchronous L3 worm cultures was prepared in Laemmli buffer with 6 M urea with glass beads in a FastPrep machine (MP Biomedicals, Irvine, CA) for 2 × 30 s. Samples were run on a 7.5% SDS-PAGE before transfer onto nitrocellulose membranes (Amersham Protran; 10600003). Membranes were blocked in TBS containing 5% milk for 1 h at RT. First antibody (Monoclonal anti-α-Tubulin clone B-5-1-2, T5168 Sigma-Aldrich 1:20,000 and rabbit polyclonal anti-PTC-3 (Eurogentec, with peptides SASHSSDDESSPAHK and EVRRGPELPKENGLG) 1:500) incubation was done O/N at 4 °C and the secondary HRP-coupled antibodies goat anti-Mouse IgG (H+L) (ThermoFisher scientific; 31430; 1:10,000) or polyclonal HRP-conjugated goat-anti-rabbit IgG (ThermoFisher scientific; 31460; 1:10,000) for 1 h at RT. The blots were developed using WesternBright ECL HRP substrate (K-12045 Advansta) in a Fusion FX7 (Vilber Lourmat) image acquisition system.

**qRT-PCR**. RNA for qRT-PCR was extracted with TRIzol according to the manufacturer's instructions from synchronous worms 26 h after L1. The RNA was DNase digested and reverse transcribed using Maxima H Minus First Strand cDNA Synthesis Kit, with dsDNase (ThermoFischer Scientific). The resulting cDNA was diluted 1:10 for further analysis. The StepOne RT-PCR system combined with StepOne Software (Applied Biotechnologies) was used for analysis. The presented values are based on three biological replicates. Expression levels were normalized to cdc-42 Primer sequences: nhr41_F, nhr41_R, nhr181_F, nhr181_R, cdc42_F, cdc42_R, nhr168_F and nhr168_R (Table S1). The data were analyzed with a two-way ANOVA followed by uncorrected Fisher's LSD test in Prism 7.

**Reporting summary**. Further information on research design is available in the Nature Research Reporting Summary linked to this article.

## Data availability

The authors declare that the data supporting the findings of this study are available within the article and source data files. Lipidomics data are available at https://lipidomes.epfl.ch/exps/2243, https://lipidomes.epfl.ch/exps/2244, and https://lipidomes.epfl.ch/exps/1709. Lipidomics de-isotoping data code is available from DOI:10.5281/zenodo.5032252. All other data supporting the findings of this study are available from the corresponding author upon request. Source data are provided with this paper.

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

## Acknowledgements
We thank J. Fürst, J. Stevens, J. Solinger, A. Stetak (Division of Molecular Neuroscience, University of Basel), I. Katic (FMI, Basel), the IMCF-Biozentrum and the Microscopy Center of the University of Zürich for the technical support, and Y. Hauser (FMI, Basel) for helpful discussions. We thank V. Solovyeva (University of Southern Denmark) for the help with the CARS experiments. S. Mango, T. Bürglin, and M. Labouesse are acknowledged for critical reading of the manuscript. We thank A. Antebi for sharing strains. Some strains were provided by the CGC, which is funded by NIH Office of Research Infrastructure Programs (P40 OD010440). The project was supported by the Swiss National Science Foundation (CRSII3_141956) (AS), (310030_184949) (HR), the NCCR Chemical Biology funded through the Swiss National Science Foundation (51NF40-185898) (HR), JSPS KAKENHI Grant Number 18K06246 (MF), and the University of Basel.

## Author contributions
A.S. and C.E.C.C. wrote the manuscript and designed the experiments. C.E.C.C. performed the majority of the experiments. J.T.H. and H.R. performed the lipidomic analysis. A.K. generated the TEM and FIB-SEM data. H.C. and M.F. generated the PTC-3-GFP *C. elegans* strain. J.B. and N.F. performed and supervised the CARS experiments. A.S., C.E.C.C., J.T.H., H.R., and A.K. analyzed the data. All authors commented on the manuscript.

## Competing interests
The authors declare no competing interests.
