## [Peer Review File · Nature Communications]

Reviewers' comments:

Reviewer #1 (Remarks to the Author):

In the present study the function of the PTCH homolog PTC-3 is investigated in *C. elegans* by iRNA knockdown. Absence of PTC-3 leads to accumulation of intracellular cholesterol, changes in ER structure and lipid droplet formation. Moreover, the authors report a reduction in acyl chain (FA) length and desaturation. This review focuses on methods used for lipid analysis and corresponding data.

It is claimed that no “major difference in the headgroup composition of the most important lipid species (Fig. 5C)” were detected. I do not agree with this statement since this figure shows a significant shift towards a higher PE fraction (about 10% increase of the molar fraction) accompanied by decreased PC and PS fractions. In my opinion this should be regarded as substantial change which may also be related to the reduction in polyunsaturated fatty acids (PUFAs) and decrease in acyl chain length and desaturation in *ptc-3*(RNAi) worms described in Fig. 5D. Data in Fig. 5D show fold-changes of the sum of glycerophospholipids. This is not very useful here, because the respective lipid class is of relevance for the biophysical properties of membranes. Therefore, I would recommend reporting changes for the individual lipid classes. Such supplementary data should contain both molar concentrations of lipid classes and species as well as species profiles (percent of the species related to total concentration of the respective lipid class).

Unfortunately, sufficient details to evaluate the methods used to quantify lipids are missing: For sterol analysis the authors refer to the following reference: Guri, Y. et al. mTORC2 Promotes Tumorigenesis via Lipid Synthesis. *Cancer Cell* 32, 807-823.e12 (2017). I could not find details on GC-MS determination of sterols in this reference. Moreover, data in Fig. 2D are shown in arbitrary units. Since changes in sterol content this is one of the key findings this has to be shown as concentrations.

Lipidomics was performed by tandem mass spectrometry using direct infusion and multiple reaction monitoring (MRM) according to the following reference: da Silveira Dos Santos, A. X. et al. Systematic lipidomic analysis of yeast protein kinase and phosphatase mutants reveals novel insights into regulation of lipid homeostasis. *Mol. Biol. Cell* 25, 3234–46 (2014). Unfortunately, this reference does not contain sufficient details concerning the methodology but refers to another reference. Because details on methodology including data processing are very important to evaluate the quality of the data the authors should either include these details or refer directly to references containing those.

Reviewer #2 (Remarks to the Author):

This is a very nice manuscript in which the authors utilize an extensive series of well-executed experiments, often using very modern techniques, to investigate the functions of PTC-3, a *C. elegans* homolog of Patched. The authors provide compelling evidence that the worm PTC-3 is likely to have cholesterol efflux property in the worm intestine and loss of this activity and subsequent

accumulation of cholesterol leads to a series of defects, many of which can be significantly countered by growing animals in cholesterol free media. They characterize the effects of loss of ptc-3 on ER morphology and identify a number of gene manipulations that form a genetic regulatory mechanism with ptc-3. The experiments are extensive, mostly very well executed, and the manuscript is well written. The manuscript is well suited for publication in this journal as it has both novel findings and likely to be of general interest.

The manuscript can benefit from a number of clarifications and minor modification:

1) Based on data presented in Figure 1C, the authors conclude that loss of ptc-3 has cell non-autonomous functions. This conclusion is most likely correct, especially given the intestinal expression pattern of this gene. However, absent controls that can definitively demonstrate that there is no ptc-3 expression and that intestine-specific RNAi strategy is definitively restricted to this tissue, the authors should moderate their claim and state the likely caveat of their results.

2) Throughout the manuscript, the authors refer to intestinal lipid/fat content as “storage”. Again, this could very well be the case but the intestine, of course, also contains lipids/fats in the form of lipoproteins such as yolk. Given that the methodologies used do not distinguish the function of the lipids/fats and that lipoproteins also arise from the ER membrane, it would be better to simply refer to lipids and fat content of the intestine rather than specify that the observed lipids function in fat storage.

3) Figure 3A-B: The authors indicate that lack of cholesterol allows ptc-3 deficient animals to reach adulthood. However, the representative image shown does not make this point very well (for example no eggs). The authors should be more clear in how they are scoring an animal as an adult.

4) The description associated with figure S2 is confusing (and perhaps misleading). Both low cholesterol and addition of 7-DHC appear to have similar effects on ptc-3 RNAi animals but the accompanying description does not make this point.

5) Figures S3, 4, S4: it would be good to state exactly which stage wild type and ptc-3RNAis were compared to each other. Were there any attempts made to have wild type animals at a stage that are roughly the same size as ptc-3 RNAi animals? It would be good to ensure that the ptc-3 RNAi phenotype is not simply a consequence of being a small sized animal.

Reviewer #3 (Remarks to the Author):

The current work by Castillo et al investigates functions of PTC-3, the homolog of mammalian PTCH-1 in *C. elegans*. They show that PTC-3 is essential for worm development. Knock-down of Ptc-3 causes cholesterol accumulation, ER morphology change and defects in lipid droplet (LD) formation. The phenotypes can be rescued by low cholesterol diet. They propose that PTC-3 transport cholesterol out of the cells. Cholesterol accumulation in cells alters ER structure and impairs LD formation. This study reveals some interesting phenotypes of Ptc-3 knock-down in *C. elegans* and suggests Ptc-3 may play a role in cholesterol metabolism. There are some questions needed to be clarified.

1. Can *C. elegans* Ptc-1 and mammalian Ptch1 rescue the phenotypes of Ptc-3 RNAi in *C. elegans*? Can mammalian PTCH-1 transport TopFluor cholesterol in yeast? The experiments in Fig. 2A should be performed with PTCH-1 since the authors assume PTCH-1 transports cholesterol. It would be very important to analyze mammalian PTCH-1 and Ptc-3 in their yeast and *C. elegans* systems side by side.
2. Mammalian NPC1 mediates cholesterol transport in lysosomes and NPC1L1 is required for intestinal cholesterol absorption. NCR-1 and NCR-2 are two homologs of NPC1 and NPC1L1 in *C. elegans*. Can the authors test Ptc-3's function in the background of Ncr-1 or Ncr-2 deficient animals? These studies might provide insights into the interplay between NCRs and PTC-3 in intracellular cholesterol transport. The results in Figure 3B suggest that cholesterol uptake is increased in Ptc-3 knock-down animals.
3. In the Ptc-3 RNAi *C. elegans*, cholesterol seems accumulate in apical membrane as illustrated by Filipin and D4H staining. The authors also propose that the ER cholesterol is increased, which then changes the ER morphology and impairs LD formation. So, the ER cholesterol content is a key and should be experimental measured.
4. LD is absent in the Ptc-3 RNAi cells. There are several possibilities including the defects in lipid absorption, the synthesis of cholesterol ester and triglyceride, or the key proteins for LD formation and structure. The authors should study the mechanism causing less LD in the Ptc-3 knock-down animals.

Detailed responses to the reviewers' comments

We wish to thank the reviewers for insightful and helpful comments, which enabled us to improve the manuscript.

Reviewer #1:

In the present study the function of the PTCH homolog PTC-3 is investigated in *C. elegans* by iRNA knockdown. Absence of PTC-3 leads to accumulation of intracellular cholesterol, changes in ER structure and lipid droplet formation. Moreover, the authors report a reduction in acyl chain (FA) length and desaturation. This review focuses on methods used for lipid analysis and corresponding data.

It is claimed that no "major difference in the headgroup composition of the most important lipid species (Fig. 5C)" were detected. I do not agree with this statement since this figure shows a significant shift towards a higher PE fraction (about 10% increase of the molar fraction) accompanied by decreased PC and PS fractions. In my opinion this should be regarded as substantial change which may also be related to the reduction in polyunsaturated fatty acids (PUFAs) and decrease in acyl chain length and desaturation in *ptc-3(RNAi)* worms described in Fig. 5D.

The increase in PE is indeed significant, and changes in PC/PE ratio are known to be associated with ER stress (Fu et al., 2011). Phosphatidylethanolamine was increased significantly, probably due to increased incorporation of shorter and more saturated fatty acids which preferentially are incorporated into phosphatidylethanolamine rather than phosphatidylcholine (Schwudke et al., 2007). In fact, loss of fatty acid desaturation affects PC/PE ratio and lipid droplet formation in *C. elegans* (Shi et al., 2013). To increase the PC production and to balance the PC/PE ratio, we fed *ptc-3(RNAi)* treated animals with 10 mM choline as previously reported to be a good concentration (Schwudke et al., 2007). Choline supplementation had no positive effect on the worm development, suggesting that the changes in PC/PE ratio are not the major cause of the developmental arrest observed in *ptc-3(RNAi)* treated animals. Because these were negative data, we did not include them into the manuscript.

Data in Fig. 5D show fold-changes of the sum of glycerophospholipids. This is not very useful here, because the respective lipid class is of relevance for the biophysical properties of membranes. Therefore, I would recommend reporting changes for the individual lipid classes. Such supplementary data should contain both molar concentrations of lipid classes and species as well as species profiles (percent of the species related to total concentration of the respective lipid class).

The complete data set is now shown in Figure S5. Molar concentrations could not be determined because the exact amount of lipid species detected varies with the repetitions.

Unfortunately, sufficient details to evaluate the methods used to quantify lipids are missing: For sterol analysis the authors refer to the following reference: Guri, Y. et al. mTORC2 Promotes Tumorigenesis via Lipid Synthesis. *Cancer Cell* 32, 807-823.e12 (2017). I could not find details on GC-MS determination of sterols in this reference.

We changed the reference to Hannich et al., 2009 . In addition, more details have been added to the methods sections. We hope we provide now enough details.

Moreover, data in Fig. 2D are shown in arbitrary units. Since changes in sterol content this is one of the key findings this has to be shown as concentrations.

The correct units for sterol analysis are “pmoles/40k adult worms” as the sterols were quantified relative to ergosterol as internal standard. This is now mentioned in the text.

Lipidomics was performed by tandem mass spectrometry using direct infusion and multiple reaction monitoring (MRM) according to the following reference: da Silveira Dos Santos, A. X. et al. Systematic lipidomic analysis of yeast protein kinase and phosphatase mutants reveals novel insights into regulation of lipid homeostasis. *Mol. Biol. Cell* 25, 3234–46 (2014). Unfortunately, this reference does not contain sufficient details concerning the methodology but refers to another reference. Because details on methodology including data processing are very important to evaluate the quality of the data the authors should either include these details or refer directly to references containing those.

A more detailed description of our lipidomics approach can be found in Guri et al. (2017) (which we reference now). In addition, more details have been added to the methods section.

Reviewer #2 (Remarks to the Author):

This is a very nice manuscript in which the authors utilize an extensive series of well-executed experiments, often using very modern techniques, to investigate the functions of PTC-3, a *C. elegans* homolog of Patched. The authors provide compelling evidence that the worm PTC-3 is likely to have cholesterol efflux property in the worm intestine and loss of this activity and subsequent accumulation of cholesterol leads to a series of defects, many of which can be significantly countered by growing animals in cholesterol free media. They characterize the effects of loss of *ptc-3* on ER morphology and identify a number of gene manipulations that form a genetic regulatory mechanism with *ptc-3*. The experiments are extensive, mostly very well executed, and the manuscript is well written. The manuscript is well suited for publication in this journal as it has both novel findings and likely to be of general interest.

Thank you!!

The manuscript can benefit from a number of clarifications and minor modification:

1) Based on data presented in Figure 1C, the authors conclude that loss of *ptc-3* has cell non-autonomous functions. This conclusion is most likely correct, especially given the intestinal expression pattern of this gene. However, absent controls that can definitively demonstrate that there is no *ptc-3* expression and that intestine-specific RNAi strategy is definitively restricted to this tissue, the authors should moderate their claim and state the likely caveat of their results.

To corroborate the notion of a potential cell non-autonomous function of PTC-3, we performed a *pos-1(RNAi)* experiment. *Pos-1* is required for embryonic pattern specification and it is active in the germline precursors. Therefore, eggs from *pos-1(RNAi)* treated animals fail to hatch (Tabara, Hill, Mello, Priess, & Kohara, 1999). L1- N2 and the intestinal specific RNAi worms (kbls7 [nhx-2p::rde-1 + rol-6(su1006)] animals) were fed with *pos-1(RNAi)* and the number of worms with alive offspring was determined. While all animals (n=28) from the intestine-specific RNAi strain gave raise to larvae i.e. alive progeny, none of the N2 worms (n=28) worms had progeny resulting in 100% embryonic lethality. These results confirm the specificity of the kbls7 [nhx-2p::rde-1 + rol-6(su1006)] strain for gut specific knockdown and therefore enforces the notion of at least one non-cell autonomous function of PTC-3. These data are included in Fig. S1C.

2) Throughout the manuscript, the authors refer to intestinal lipid/fat content as “storage”. Again, this could very well be the case but the intestine, of course, also contains lipids/fats in the form of lipoproteins such as yolk. Given that the methodologies used do not distinguish the function of the lipids/fats and that lipoproteins also arise from the ER membrane, it would be better to simply refer to lipids and fat content of the intestine rather than specify that the observed lipids function in fat storage.

To address this point, we measured yolk content on oocytes, which endocytose yolk synthesized in the intestine. We did not observe any significant difference in yolk uptake when we compared mock- and *ptc-3(RNAi)* treated animals, consistent with the notion of a fat storage defect. These data are now included in Fig. S1E. Nevertheless, except at the end of the paragraph in which we mention the experiment and we offer fat storage problems as a possibility, we refer throughout the text to ‘fat content’ rather than ‘fat storage’.

3) Figure 3A-B: The authors indicate that lack of cholesterol allows *ptc-3* deficient animals to reach adulthood. However, the representative image shown does not make this point very well (for example no eggs). The authors should be more clear in how they are scoring an animal as an adult.

In general, low cholesterol rescued animals are smaller than mock treated animals. The animals do contain eggs, which can be observed at a higher magnification. The figure has been updated to show a higher magnification where eggs or oocytes can be observed

4) The description associated with figure S2 is confusing (and perhaps misleading). Both low cholesterol and addition of 7-DHC appear to have similar effects on *ptc-3* RNAi animals but the accompanying description does not make this point.

7-DHC plates do not contain cholesterol. Therefore, our conclusion is that the addition of 7-DHC does not have an effect on top of cholesterol depletion. We modified the text to clarify this point.

5) Figures S3, 4, S4: it would be good to state exactly which stage wild type and *ptc-3* RNAis were compared to each other. Were there any attempts made to have wild type animals at a stage that are roughly the same size as *ptc-3* RNAi animals? It would be good to ensure that the *ptc-3* RNAi phenotype is not simply a consequence of being a small sized animal.

For comparison between Mock and *ptc-3(RNAi)*, worms were fed with *ptc-3(RNAi)* after L2 stage, consequently they were able to reach adulthood, we have now specified in the figure legends when this feeding scheme was used.

Reviewer #3 (Remarks to the Author):

The current work by Castillo et al investigates functions of PTC-3, the homolog of mammalian PTCH-1 in *C. elegans*. They show that PTC-3 is essential for worm development. Knock-down of Ptc-3 causes cholesterol accumulation, ER morphology change and defects in lipid droplet (LD) formation. The phenotypes can be rescued by low cholesterol diet. They propose that PTC-3 transport cholesterol out of the cells. Cholesterol accumulation in cells alters ER structure and impairs LD formation. This study reveals some interesting phenotypes of Ptc-3 knock-down in *C. elegans* and suggests Ptc-3 may play a role in cholesterol metabolism. There are some questions needed to be clarified.

1. Can *C. elegans* Ptc-1 and mammalian Ptch1 rescue the phenotypes of Ptc-3 RNAi in *C. elegans*?

We generated a *C. elegans* strain, which would express mammalian PTCH1. When we performed *ptc-3(RNAi)* on these worms, we could not observe any rescue. However, western blot analyses demonstrated that *ptc-3(RNAi)* also led to downregulation of PTCH1. Hence the PTCH1 levels might be too low to rescue *ptc-3(RNAi)* phenotypes. This was quite

surprising, but very reproducible. A possible explanation is that one or more of the siRNAs generated from the about 2 kb original construct by the feeding bacteria would also target the PTCH1 RNA.

Can mammalian PTCH-1 transport TopFluor cholesterol in yeast? The experiments in Fig. 2A should be performed with PTCH-1 since the authors assume PTCH-1 transports cholesterol. It would be very important to analyze mammalian PTCH-1 and Ptc-3 in their yeast and *C. elegans* systems side by side.

The ability of PTCH1 to transport cholesterol in yeast was already reported in Bidet et. 2011. We expressed PTCH1 and PTC-3 in HEK293. In both cases we observed a significant increase of TopFluor cholesterol efflux compared to mock treated cells, further demonstrating that PTCH1 and PTC-3 are cholesterol permeases.

2. Mammalian NPC1 mediates cholesterol transport in lysosomes and NPC1L1 is required for intestinal cholesterol absorption. NCR-1 and NCR-2 are two homologs of NPC1 and NPC1L1 in *C. elegans*. Can the authors test Ptc-3's function in the background of Ncr-1 or Ncr-2 deficient animals? These studies might provide insights into the interplay between NCRs and PTC-3 in intracellular cholesterol transport. The results in Figure 3B suggest that cholesterol uptake is increased in Ptc-3 knock-down animals.

This is a very nice idea! We performed a triple knockdown of *ncr-1*, *ncr-2* and *ptc-3*. Even though we observed a slight increase in development when compared to the *ptc-3* single knockdown, this was not statistically significant. This lack of significance is probably due to large data spread, which is unfortunately not uncommon in triple knockdown experiments. These data are shown in Fig. 3E.

3. In the Ptc-3 RNAi *C. elegans*, cholesterol seems accumulate in apical membrane as illustrated by Filipin and D4H staining. The authors also propose that the ER cholesterol is increased, which then changes the ER morphology and impairs LD formation. So, the ER cholesterol content is a key and should be experimental measured.

To measure the intracellular distribution of cholesterol is incredibly challenging. As already indicated by the filipin and DH4 images provided in the initial manuscript, the resolution is not good enough to unambiguously detect the cholesterol in the ER. The filipin is bleaching so quickly, that we cannot image for long enough, and the DH4-mCherry is overexpressed in the cytoplasm, making it impossible to detect intracellular membrane recruitment. Given the lack of suitable methods to demonstrate cholesterol accumulations in the ER by live cell imaging, we attempted to detect cholesterol by CARS microscopy. We would like to stress that this has never been attempted in *C. elegans*. For this, we fed worms with either deuterated cholesterol or ¹³C-labelled cholesterol before CARS imaging. Unfortunately, neither D-cholesterol nor ¹³C-labelled cholesterol accumulated at high enough

concentrations intracellularly that we could confidently detect them. We are not aware of any other method we could potentially use to fulfil the reviewer's request. Thus, with the current methods available, the desired experiment cannot be performed.

4. LD is absent in the *Ptc-3* RNAi cells. There are several possibilities including the defects in lipid absorption, the synthesis of cholesterol ester and triglyceride, or the key proteins for LD formation and structure. The authors should study the mechanism causing less LD in the *Ptc-3* knock-down animals.

We explored the different avenues to provide more mechanistic insight in the LD phenotype as suggested by the reviewer. It is unlikely that defects in cholesterol ester synthesis contribute to the LD phenotype since in *C. elegans*, as very little if any cholesterol or cholesterol esters are present in LDs in *C. elegans* (Vrablik et al. 2015, Lee et al., 2015). To explore whether defects in lipid absorption could be the reason for the lack of LDs, we check for signs of increased autophagy, both in EM and using the *C. elegans* LC3 LGG-1::GFP as a marker in live cell imaging. However, we observed no sign of increased autophagy, suggesting that the nutritional status is fine and lipid absorption are not defective. The data on LGG-1::GFP are shown in Fig.S3A. Next, we tested whether LDs are absent due to an increase of lipolysis. To this end, we determined the level of the triglyceride lipase ATGL-1 in *ptc-3(RNAi)* worms. Again, we did not detect any difference when compared to mock treated animals, indicating that the absence of LD is not due to an increase in ATGL-1. These data are shown in Fig. S3A. Finally, we checked the levels of DHS-3, a component of lipid droplets. We did not detect any significant difference in the DHS-3 protein levels in *ptc-3(RNAi)* worms compared to mock treated animals. These data are shown in Fig. S3B. All these results, albeit negative, are in support of our model that the membrane of the endoplasmic reticulum is too stiff -potentially through accumulation of cholesterol- to form lipid droplets in the absence of *ptc-3 (RNAi)* animals.

REVIEWER COMMENTS

Reviewer #1 (Remarks to the Author):

In revised manuscript, the authors added lipid data and some clarification concerning the lipidomics method. However, I have concerns regarding the quality of lipidomic data (specified below) and the conclusions derived from these data:

- Unfortunately, the references concerning direct infusion lipidomics are still the same as in the previous version. Despite more details from these references were added important information is still missing e.g. data processing like deisotoping of Type II overlap (= overlap resulting mainly from ¹³C-atoms in double bond series) which is required for accurate identification and quantification in direct analysis. Without this type of correction data are incorrect due to substantial isotopic overlap.
- Lipid extraction: The phase separation performed does match neither the published protocol nor that of Bligh and Dyer – “phase separation was induced by addition of 0.5 mL MS-H₂O and 0.5 ml CHCl₃” the published protocol (Schwudke, D. et al. Anal. Chem. (2007) 79, 4083–4093) used for the same volumes 1ml each to induce phase separation (which matches the ratios described by Bligh and Dyer). My guess is that phase separation using only half volume results in poor phase separation.
- To evaluate the effect of PCT-3 knockdown on the membrane physiology, it is important to include cholesterol as a fraction into the calculations e.g. in Fig. 5C – what is the molar ratio of cholesterol to phospho-/sphingolipids? Related to this I do not understand the reply of the authors to Fig. 5D that “Molar concentrations could not be determined because the exact amount of lipid species detected varies with the repetitions.”. Figure 5C is also based on molar concentrations – which are calculated from lipid species data. Such data were also added as supplementary data but lack clarity because numerous species are listed which are most likely below LOD – please remove such species from the tables and include mean and SD. Moreover, there are huge variations e.g. for PI species – please check or repeat these experiments which are implausible (e.g. PI(O⁻)₃₆:0 was present at 15% or practically absent in PCT-3 KO).
- To explain the phenotype of PCT-3 KO, the authors propose that membrane physics with decreased membrane fluidity could provide an explanation (see comments above - integration of sterol content into calculation is mandatory). Further, to rule out that the change in PC/PE is not an underlying cause, choline feeding was applied without positive effect on the worm development. Did choline feeding increase the PC content? Please add these data. Moreover, I would strongly recommend adding lipidomic data of the NHR-49 and FAT-7 overexpression in ptc-3(RNAi) animals. Changes in the lipidome should substantiate the concept of membrane fluidity as potential explanation of the phenotype.
- Annotation of CL species does not fit to current standards for annotation e.g. CL_{68:4_C16:1}. Most likely, this means that a 16:1 acyl chain is present in this CL species - proper annotation would be CL 16:1_52:3. GlcCer should be renamed to HexCer unless the the hexose is identified.

Reviewer #2 (Remarks to the Author):

The authors have addressed all of my concerns. This is a very nice manuscript!

Reviewer #3 (Remarks to the Author):

The authors have done more work to address the reviewer's questions. This ms has been improved and I recommend to accept.

We wish to thank Reviewers #2 and #3 for their positive assessment and for recommending publication of our work.

Reviewer #1 still had some concerns, which we have taken seriously and provide comments below. Where necessary, we introduced changes into manuscript.

Reviewer #1 (Remarks to the Author):

In revised manuscript, the authors added lipid data and some clarification concerning the lipidomics method. However, I have concerns regarding the quality of lipidomic data (specified below) and the conclusions derived from these data:

Unfortunately, the references concerning direct infusion lipidomics are still the same as in the previous version. Despite more details from these references were added important information is still missing e.g. data processing like deisotoping of Type II overlap (= overlap resulting mainly from ¹³C-atoms in double bond series) which is required for accurate identification and quantification in direct analysis. Without this type of correction data are incorrect due to substantial isotopic overlap.

We did not perform any de-isotoping. We clarify this in the methods section now. We chose not to do the de-isotoping because this also introduces additional errors as any kind of data manipulation. We agree that de-isotoping is essential when characterizing a lipidome in depth in order to correctly quantify every lipid species and its double bond structure. We are not claiming to present a quantitative lipidome of worms. We are, on the other hand, comparing two conditions and looking for differences between these conditions. In this case, de-isotoping does not have an impact on the comparison; actually, quite the opposite because it would, as outlined above, introduce errors. We would like to point out that the raw, unaltered data will be openly available, and hence de-isotoping could be performed by any interested party.

Lipid extraction: The phase separation performed does match neither the published protocol nor that of Bligh and Dyer – “phase separation was induced by addition of 0.5 mL MS-H₂O and 0.5 ml CHCl₃” the published protocol (Schwudke, D. et al. Anal. Chem. (2007) 79, 4083–4093) used for the same volumes 1ml each to induce phase separation (which matches the ratios described by Bligh and Dyer). My guess is that phase separation using only half volume results in poor phase separation.

Phase separation depends on the solvent mixtures and on the matrix of the extracted material. This needs to be optimized for each matrix. For *C. elegans* extracts, addition of 0.5 ml MS-H₂O is sufficient to induce robust phase separation. The 0.5 ml CHCl₃ is added to facilitate the collection of the organic phase. The procedure is described in detail in the materials and methods, and researchers can follow this protocol to obtain robust phase separations in lipid extractions from *C. elegans*. As far as the references are concerned, we added in the text ‘with minor modifications’ to clarify this point. At any rate, we describe in detail the extraction in the section. We hope this change satisfies the reviewer’s concern.

To evaluate the effect of PCT-3 knockdown on the membrane physiology, it is important to include cholesterol as a fraction into the calculations e.g. in Fig. 5C – what is the molar ratio of cholesterol to phospho-/sphingolipids?

Unfortunately, we are unable to directly compare the molar ratio of cholesterol to phospho-/sphingolipids. *C. elegans* has relative low concentrations of sterols. Therefore, to determine the relative sterol levels, we used 40,000 animals per biological replicate. In contrast, for the phospho-

/sphingolipid analysis only 8,000 animals were used in independent biological replicates. Since the lipids were not extracted at the same time/nor from the same batch, we do not feel confident to directly compare sterol to phospholipid levels. We agree that our description may not have been precise enough and we amended this part.

We would also like to stress, that we used different methods, including lipidomics, to show that cholesterol is intracellularly enriched in the absence of PTC-3. The imaging techniques allowed us clearly to pinpoint a major change in a specific organ, the intestine. In contrast, the lipidomics were of course performed on entire animals. It is conceivable and likely, that not all organs are affected to the same extent, dependent on whether or not PTC-3 is expressed in a certain tissue. Therefore, even if we could determine the precise ratio of cholesterol to phospho-/sphingolipids, it would be misleading, since PTC-3 is not equally expressed in all tissues or rather absent in some tissues.

Related to this I do not understand the reply of the authors to Fig. 5D that “Molar concentrations could not be determined because the exact amount of lipid species detected varies with the repetitions.”. Figure 5C is also based on molar concentrations – which are calculated from lipid species data.

While total amounts of lipids were not significantly different between mock and *ptc-3 (RNAi)* treatment there was quite a spread within the samples. To make the individual biological replicates more comparable, we decided to calculate the Mol % for individual lipid species. Of course, molar concentrations can in principle be calculated, and we included an example for PC lipid species below. However, as outlined out above, we feel that this is not a useful representation of our data in particular with respect to sterols. Absolute values/concentrations would be misleading in our view, and therefore we prefer to keep the relative levels. Since all the data are available, if somebody is really interested in the molar concentration, this information can be extracted from the data.

available, if somebody is really interested in the molar concentration, this information can be extracted from the data.

In figure 5D the fold change value between the Mock and *ptc-3(RNAi)* for each individual lipid was calculated. To avoid introducing errors due to big changes generated by molecules with very low abundance, only species with an original value above 0.01 were considered. From these values the average and SEM were calculated, and all the values for species with 34 to 38 carbons were plotted. The detailed lipid distributions can be seen in the supplementary figure S5A to S5E.

Such data were also added as supplementary data but lack clarity because numerous species are listed which are most likely below LOD – please remove such species from the tables and include mean and SD. Moreover, there are huge variations e.g. for PI species – please check or repeat these experiments which are implausible (e.g. PI(O-)36:0 was present at 15% or practically absent in PCT-3 KO).

Indeed, PI species have very low concentrations in *C. elegans* (less than 5 Mol%, see Fig. 5C). This leads to greater variability in these lipid measurements. Lipid species are reported according to the LOD of the corresponding internal standard for each lipid class as described in the Methods section and references therein. Figure S5C shows % within the lipid class as requested by the Reviewer. For minor species of minor lipid classes like PI small changes in concentration can lead to larger variations in this kind of data representation. This is no surprise to any kind of lipidomics specialist interested in this supplementary data. Rather than removing this data, we chose to show it with individual data points, mean values and error bars as described in Figure S5C figure. We would also like to point out that we only plotted species that represent more than 1% of the total of an individual lipid class.

To explain the phenotype of PCT-3 KO, the authors propose that membrane physics with decreased membrane fluidity could provide an explanation (see comments above - integration of sterol content into calculation is mandatory). Further, to rule out that the change in PC/PE is not an underlying cause, choline feeding was applied without positive effect on the worm development. Did choline feeding increase the PC content? Please add these data. Moreover, I would strongly recommend adding lipidomic data of the NHR-49 and FAT-7 overexpression in *ptc-3(RNAi)* animals. Changes in the lipidome should substantiate the concept of membrane fluidity as potential explanation of the phenotype.

The data on NHR-49 and FAT-7 were already present in the initial version of the manuscript. The reviewer did not ask for additional lipidomics data in the first round of review. It is unjustified and unfair to ask about these additional data in the second round. Per request of the editor, we added a sentence in the discussion: 'How the entire lipidome is affected under these conditions remains to be determined.'

We are not the first to use choline feeding to increase phosphatidylcholine levels in *C. elegans*. Besides the Schwudke et al. (2007) paper that we cited in our manuscript, other examples include Brendza et al. *Biochem. J.* (2007), Li et al., *J. Biochem* (2011), Klapper et al., *Genes & Nutrition* (2016). This is not an exhaustive list. We would also like to point out that the reduction in PC content appears to increase lipogenesis and lipid droplet size in *C. elegans* (Walker et al., *Cell* 2011, Ehmke et al., *Genes & Nutrition* 2014). We observe, however, the absence of LDs, which is a different phenotype altogether. Since, we did not observe a positive effect of choline feeding on the *ptc-3(RNAi)* phenotype, and the reported phenotype of reduced PC levels appears to be different from the *ptc-3(RNAi)* phenotype, we consider it non-essential to perform lipidomics on choline fed animals.

Annotation of CL species does not fit to current standards for annotation e.g. CL68:4_C16:1. Most likely, this means that a 16:1 acyl chain is present in this CL species - proper annotation would be CL 16:1_52:3. GlcCer should be renamed to HexCer unless the hexose is identified.

Annotation of CL species followed the logic of the MRM measurement method. It shows the composition of the intact molecule (CL68:4) as selected in the first quadrupole and the detected fragment (_C16:1) as selected in the third quadrupole of the mass spectrometer. Still, to concede to the reviewer's request, we changed the nomenclature according to her/his suggestion.

Unlike in mammals, in which both glucose and galactose are incorporated into hexosylceramides, glucose is the only hexose found in hexosylceramides of *C. elegans* (Chitwood et al., Lipids 1995).

REVIEWER COMMENTS

Reviewer #1 (Remarks to the Author):

In second revised version of the manuscript, the authors added only some more details concerning the lipidomics method. Unfortunately, the clarification justified the concerns concerning inappropriate lipidomic analysis raised by this reviewer.

- Concerning Type-II isotopic overlap, the authors explained that de-isotoping was not performed because “Any kind of data manipulation always introduces also new errors which can even be counterproductive when comparing conditions.” Such statement is misleading because without correction of these overlap the data are simply wrong. De-isotoping is not data-manipulation but an essential step to generate correct data! This is mandatory in any shotgun lipidomics workflow except ultra-high mass resolution resolves overlapping isobaric peaks. Without this step the reported Mol% are also incorrect – for example the M+2 isotopic peak of PI 38:4 is about 7% related to the monoisotopic peak; data in Fig. 5D report about 7% PI 38:3 which are originating mainly from the M+2 of PI 38:4.

- Data used for calculation of fold-change in Fig. 5D e.g. for 34:0 are based on a minor fraction lipidome fraction (0.2% Mock vs. 0.4% ptc3 of all species used for calculation). Considering that, these data contain also M+2 isotopic overlap resulting from 34:1 the reported values are also too high. Moreover, substantial fold-changes shown e.g. for C36:0 (0.3% Mock vs. 0.4% ptc3) and monounsaturated C38:1 (0.3% Mock vs. 0.4% ptc3) are only negligible fractions. Thus, the fold-changes shown in Fig. 5D may be misleading because it is not obvious which lipid fraction is affected by these changes. This is important because finally, lipid composition is key to get insight into biophysics of membranes and this should reflect the quantity of lipid species.

Here it is also unclear to me, why ether/odd-chain-species for PC and PE were not included into calculations despite such species represent major species e.g. PE33:1/PE(O-)34:1 (Fig. 5S). In contrast, for PS and PI ether/odd-chain-species were included. Is there any reason for this data selection? The lipidomic approach is not able to differentiate ether and odd chain species – so I would recommend annotating species based on knowledge (odd chain species could also indicate the presence of branched chain acyl chains; please indicate when assumption are applied). In general, I would recommend to perform a separate calculation for ether species, if their presence in the worms is justified by literature. When the literature provides evidence for odd chain species they should be added into calculation for the respective lipid classes. Are these changes statistical significant because the Fig. legend describes “There is a shift from PUFAs to saturated FA and MUFAs.”?

- Concerning lipid extraction: “Phase separation depends on the solvent mixtures and on the matrix of the extracted material. This needs to be optimized for each matrix.” Yes, I fully agree with this statement especially when existing protocols are changed. Therefore, I would like to ask the authors to provide data concerning on the optimization of *C. elegans* lipids. This is required because I do not agree that using only half volume of water and chloroform (compared to Bligh & Dyer and Schwudke et al.) could be considered as “minor modification”. Solvent composition is critical for effective lipid recovery.

- The data contain numerous species close or below LOD – the authors did not remove such species as requested. As pointed out above this may lead to wrong interpretation. Such kind of filtering could also not be done by interested researches because it is necessary to have insight into the raw data including for example extractions blanks – this kind of data validation has to be performed by the authors to avoid misinterpretation caused by reporting of species <LOD. For example, PI comprises 123 species and only about 10-15 species have an abundance above 1% - presumably

about 100 species seem to be negligible and many of them are <LOD.

- Annotation of cardiolipin species was not changed in the suppl. table.
- I do not understand why the authors keep refusing to calculate and discuss phospholipid to sterol ratios. May be it is related to the minor fraction of sterols? One argument of the authors is that lipid changes affect only the intestine. If so, this findings could be discussed. To show and discuss such data is scientific but not misleading as suggested by the authors. Moreover, it is not true that interested reader may calculate sterol/phospholipid ratios because sterols were presented as pmol/40K worms while other lipid data are related to phosphorus content – the unit for suppl. data for Fig. 5C pmol/? is missing! Reporting of transparent lipid data is essential to support also the model presented in Fig. 7 which links membrane biophysics including cholesterol to ER structure and LD formation.

In summary, the response of the authors does not fit to their statement that the raised concerns were “taken seriously”. Most of the issues raised by this reviewer were not addressed appropriately. As explained above, data and method are in part unsound. Moreover, data presentation is misleading (e.g. concerning Fig. 5D or calculation of sterol/phospholipid ratios) and lacks thorough quality control.

Point-by-point response to the reviewer's concerns:

Reviewer #1 (Remarks to the Author):

In second revised version of the manuscript, the authors added only some more details concerning the lipidomics method. Unfortunately, the clarification justified the concerns concerning inappropriate lipidomic analysis raised by this reviewer.

We do not agree with this assessment. Our analysis is not inappropriate, it is simply not what the reviewer seems to want. The reviewer seems to want us to claim we have measured absolute amounts of the different lipid species, but this is not what we have claimed. We have measured relative amounts and changes between mutant and wild type. Since the original, raw data will be released upon publication, everybody is free to analyze the data the way s/he likes.

- Concerning Type-II isotopic overlap, the authors explained that de-isotoping was not performed because “Any kind of data manipulation always introduces also new errors which can even be counterproductive when comparing conditions.” Such statement is misleading because without correction of these overlap the data are simply wrong. De-isotoping is not data-manipulation but an essential step to generate correct data! This is mandatory in any shotgun lipidomics workflow except ultra-high mass resolution resolves overlapping isobaric peaks. Without this step the reported Mol% are also incorrect – for example the M+2 isotopic peak of PI 38:4 is about 7% related to the monoisotopic peak; data in Fig. 5D report about 7% PI 38:3 which are originating mainly from the M+2 of PI 38:4.

De-isotoping is not the standard analysis for MRM data. If it were, there would be an automated tool that would accept MRM data, but these are not available. So, it is impossible to understand, why this type of analysis is mandatory, in particular because when comparing mutant to wild type almost all of the effects of these corrections cancel out mathematically. To fulfill the reviewers request, we had to do it manually for all lipid species. After a lot of unnecessary work, we are happy to report that the de-isotoping did not change anything about the outcome of the analysis, and consequently it did not change any interpretation of the lipidomics data.

- Data used for calculation of fold-change in Fig. 5D e.g. for 34:0 are based on a minor fraction lipidome fraction (0.2% Mock vs. 0.4% ptc3 of all species used for calculation). Considering that, these data contain also M+2 isotopic overlap resulting from 34:1 the reported values are also too high. Moreover, substantial fold-changes shown e.g. for C36:0 (0.3% Mock vs. 0.4% ptc3) and monounsaturated C38:1 (0.3% Mock vs. 0.4% ptc3) are only negligible fractions. Thus, the fold-changes shown in Fig. 5D may be misleading because it is not obvious which lipid fraction is affected by these changes. This is important because finally, lipid composition is key to get insight into biophysics of membranes and this should reflect the quantity of lipid species.

We respectfully disagree with the reviewer's assessment. The meaningfulness of the lipidomics data has been shown by our findings presented in Figure 6 that overexpression of

the desaturase FAT-7 and modulating the nuclear hormone receptors, which controlling the expression of desaturases rescue the *ptc-3(RNAi)* phenotypes.

Here it is also unclear to me, why ether/odd-chain-species for PC and PE were not included into calculations despite such species represent major species e.g. PE33:1/PE(O-)34:1 (Fig. 5S). In contrast, for PS and PI ether/odd-chain-species were included. Is there any reason for this data selection? The lipidomic approach is not able to differentiate ether and odd chain species – so I would recommend annotating species based on knowledge (odd chain species could also indicate the presence of branched chain acyl chains; please indicate when assumption are applied). In general, I would recommend to perform a separate calculation for ether species, if their presence in the worms is justified by literature. When the literature provides evidence for odd chain species they should be added into calculation for the respective lipid classes. Are these changes statistical significant because the Fig. legend describes “There is a shift from PUFAs to saturated FA and MUFAs.”?

We agree with the reviewer that we cannot distinguish between odd chain and ether lipid species. This is now clearly marked in the supplemental data file. However, because their nature is ambiguous, we did not include them into the analysis.

- Concerning lipid extraction: “Phase separation depends on the solvent mixtures and on the matrix of the extracted material. This needs to be optimized for each matrix.” Yes, I fully agree with this statement especially when existing protocols are changed. Therefore, I would like to ask the authors to provide data concerning on the optimization of *C. elegans* lipids. This is required because I do not agree that using only half volume of water and chloroform (compared to Bligh & Dyer and Schwudke et al.) could be considered as “minor modification”. Solvent composition is critical for effective lipid recovery.

We maintain that the requested experiment is not necessary. We processed the mock control and RNAi samples side-by-side and hence even if we did not extract 100% of the lipids, the extent to which we extract the lipids are the same for the different samples. Internal standards were spiked into the extractions that correct for differences in extraction efficiency. Furthermore, we find this request unwarranted for basic biochemistry reasons. There is no optimal lipid extraction. The different lipid species are extracted best under different conditions. Highly polar lipids are not extracted best under the same conditions as neutral lipids. Therefore, optimization is something that can be done for a particular lipid class or for a particular experiment, but is not something general. There is no benefit to our manuscript, nor to the scientific literature in doing the experiments suggested by the reviewer.

As the sterol extraction and the phospholipid extraction follow two different protocols, optimized for each lipid species, we can also not directly compare them; they lack a common baseline to which we could normalize them to.

- The data contain numerous species close or below LOD – the authors did not remove such species as requested. As pointed out above this may lead to wrong interpretation. Such kind of filtering could also not be done by interested researches because it is necessary to have insight into the raw data including for example extractions blanks – this kind of data

validation has to be performed by the authors to avoid misinterpretation caused by reporting of species <LOD. For example, PI comprises 123 species and only about 10-15 species have an abundance above 1% - presumably about 100 species seem to be negligible and many of them are <LOD.

We respectfully disagree with the reviewer. To please the reviewer, we applied filtering, which falsified the outcome of the experiments. MUFAs are very low abundant lipid species, which would be removed by the suggested filtering mechanism. In the *ptc-3(RNAi)*, we observe a drastic increase in these lipid species. By filtering, we would exclude MUFAs from the analysis, thereby preventing the discovery that they actually increase dramatically in the *ptc-3(RNAi)* animals. If we were to follow the reviewer's advice, we would misinterpret our data. We fail to understand why the reviewer insists that we should be less rigorous in our data analysis and why s/he continues to request that we should misinterpret our data.

- Annotation of cardiolipin species was not changed in the suppl. table.

We do not understand this complaint, as we have changed the cardiolipin species, and they are also changed in the source data.

- I do not understand why the authors keep refusing to calculate and discuss phospholipid to sterol ratios. May be it is related to the minor fraction of sterols? One argument of the authors is that lipid changes affect only the intestine. If so, this findings could be discussed. To show and discuss such data is scientific but not misleading as suggested by the authors. Moreover, it is not true that interested reader may calculate sterol/phospholipid ratios because sterols were presented as pmol/40K worms while other lipid data are related to phosphorus content – the unit for suppl. data for Fig. 5C pmol/? is missing! Reporting of transparent lipid data is essential to support also the model presented in Fig. 7 which links membrane biophysics including cholesterol to ER structure and LD formation.

We fail to understand why the reviewer insists that we compare apples with pears. We are very transparent in the way we report our lipid data. As we pointed out previously, for the sterol analyses we needed 5 times the amount of worms for extraction compared to the phospholipid analysis because the sterol levels are very low. Second, to have optimal extraction conditions, the sterol extraction protocol is different than that for phospholipids. So, the amount of biological material as well as the extraction methods are not the same. Yet, the reviewer demands repeatedly that we compare the sterols levels and the phospholipids directly. Even if we would use inorganic phosphate for normalization, the extraction levels of phospholipids in the two methods are different because they were optimized for one or the other lipid species. Therefore, we might misrepresent the sterol levels. We prefer to compare things that are comparable, but normalizing sterol levels to inorganic phosphate would be wrong and therefore we are still not going to do this. Unlike what the reviewer claims, this does not at all interfere with our model. We show by different means that cholesterol levels are increased in *ptc-3(RNAi)* animals, that this is detrimental for the worms and that a no cholesterol diet alleviates this phenotype i.e. LD formation. The model does not rest only on a particular experiment but rather by

addressing the same point by different experimental means. For more comments, see also the point on the lipid extraction method.

In summary, the response of the authors does not fit to their statement that the raised concerns were “taken seriously”. Most of the issues raised by this reviewer were not addressed appropriately. As explained above, data and method are in part unsound. Moreover, data presentation is misleading (e.g. concerning Fig. 5D or calculation of sterol/phospholipid ratios) and lacks thorough quality control.

We respectfully disagree. We have taken the concerns seriously and responded to each of them in the last version of the manuscript and the point-by-point response.

REVIEWER COMMENTS

Reviewer #4 (Remarks to the Author):

The authors should provide additional information on the lipidomics annotation scheme as well as the mass spec instrumental parameters employed for fragmentation of each lipid class.

The authors should include the internal standards incorporated for each lipid class.

This work does not mention the use of quality controls. If these were used, how were they incorporated to ensure data quality. What was the %CV of the lower level lipid species reported in this work based on QC data.

The authors should provide literature for the modification performed with their lipid extraction (i.e., has this lipid extraction as performed by the authors been published elsewhere?). Otherwise, the authors should remove terminology that references to a previously published lipid extraction such as the Bligh- Dyer.

Responses to the reviewer's comments to the authors:

The authors should provide additional information on the lipidomics annotation scheme as well as the mass spec instrumental parameters employed for fragmentation of each lipid class.

In the Materials and Methods section we provide the m/z window for the precursor ions and the class specific transitions used for lipid annotations and quantifications.

The authors should include the internal standards incorporated for each lipid class.

We used internal standards as already outlined in the manuscript (p. 19): Lysates were eluted into glass tubes with lipid standards (glycerophospholipid and sphingolipid standards: di-lauryl phosphatidylcholine, di-lauryl phosphatidylethanolamine, di-lauryl phosphatidylinositol, di-lauryl phosphatidylserine, tetra-lauryl cardiolipin, C17 ceramide, C12 sphingomyelin, C8 glucosylceramide, all from Avanti Polar Lipids; sterol standard: ergosterol from Fluka) and beads were washed and eluted again with 200 μ l MS-H₂O.

This work does not mention the use of quality controls. If these were used, how were they incorporated to ensure data quality. What was the %CV of the lower level lipid species reported in this work based on QC data.

No additional QC samples were included in the lipidomics analysis. Lower limit of quantification was determined based on the linearity of the standard curves recorded for the internal standards.

The authors should provide literature for the modification performed with their lipid extraction (i.e., has this lipid extraction as performed by the authors been published elsewhere?). Otherwise, the authors should remove terminology that references to a previously published lipid extraction such as the Bligh- Dyer.

We removed the Bligh-Dyer reference and cite instead: Hannich, J. T. et al. 1-Deoxydihydroceramide causes anoxic death by impairing chaperonin-mediated protein folding. Nat. Metab. 1, 996–1008 (2019).

Responses to the comments in the reviewer's file.

- Concerning Type-II isotopic overlap, the authors explained that de-isotoping was not performed because “Any kind of data manipulation always introduces also new errors which can even be counterproductive when comparing conditions.” Such statement is misleading because without correction of these overlap the data are simply wrong.

Deisotoping

is not data-manipulation but an essential step to generate correct data! This is mandatory in any shotgun lipidomics workflow except ultra-high mass resolution resolves overlapping isobaric peaks. Without this step the reported Mol% are also incorrect – for example the M+2 isotopic peak of PI 38:4 is about 7% related to the monoisotopic peak; data in Fig. 5D report about 7% PI 38:3 which are originating mainly from the M+2 of PI 38:4.

The reviewer is correct that with shotgun lipidomics there may be isobaric overlaps between the M+2 isotope and the monoisotopic peak of a lipid w/ one less double bond with full scan data. This becomes even more profound if a high-resolution mass spectrometer is not used. The authors mention, however, that they collected data using MRM modes. Therefore, there was most likely a tight window set around the precursor of interest. The authors are monitoring precursor and fragment ions to ensure certainty in the

assignment. Therefore, deisotoping isn't as much of a concern with MRM. With that being said, the authors should make sure that the isolation window used for MRM is mentioned and how the lipid annotations were performed (i.e., how was certainty in the lipid assignment ensured, software used for lipid annotation, etc.) It's unclear from the conversation between the reviewer and the author how M+2 data was used or how that information was extracted from the mass spectra. Traditionally, the adduct of the precursor (e.g., [M+H]⁺, [M+Na]⁺, [M+NH₄]⁺) is used for quantitation purposes.

We thank reviewer 4 for agreeing that de-isotoping is not an issue with MRM. We provided in the last version of the manuscript a description on how we performed the de-isotoping. We now provide more information in the Material and Methods section on the transitions, the precursors, and the precursor mass window that was the basis for our annotations and quantifications.

The collision energy was optimized for each lipid class based on the internal standards and the following m/z transitions were measured using an m/z window of +/- 0.5 amu for the precursors: in positive electron spray ionization mode (ESI⁺) phosphatidylcholine M+H⁺ (Q1) -> 184.07 (Q3), phosphatidylethanolamine M+H⁺ (Q1) -> neutral loss of 141.02 (Q3), in negative electron spray ionization mode (ESI⁻) phosphatidylinositol M-H⁺ (Q1)->241.01 (Q3), phosphatidylserine M-H⁺ (Q1)->neutral loss of 87.03 (Q3) and cardiolipin M-2H+2 (Q1)-> different fatty acid fragments (Q3).

- Data used for calculation of fold-change in Fig. 5D e.g. for 34:0 are based on a minor fraction lipidome fraction (0.2% Mock vs. 0.4% ptc3 of all species used for calculation). Considering that, these data contain also M+2 isotopic overlap resulting from 34:1 the reported values are also too high. Moreover, substantial fold-changes shown e.g. for C36:0 (0.3% Mock vs. 0.4% ptc3) and monounsaturated C38:1 (0.3% Mock vs. 0.4% ptc3) are only negligible fractions. Thus, the fold-changes shown in Fig. 5D may be misleading because it is not obvious which lipid fraction is affected by these changes. This is important because finally, lipid composition is key to get insight into biophysics of membranes and this should reflect the quantity of lipid species.

This statement appears to be a continuation of the concern from the reviewer in the first bullet point. Once again, I'm not sure how M+2 data was used by the authors. The authors should ensure that a thorough description of the "lipidome fraction" is provided in the manuscript. The authors present sum composition of the lipid species, so it appears that they are reporting based on certainty in the lipid annotations as obtained from fragmentation data.

Lipidomics data were obtained based on MRM analysis using lipid class specific transitions as described in detail in the Materials and Methods section. Supplementary Figure 5 shows the distribution of lipid species for each lipid class for interested readers who would like to see more detailed information than the fold changes shown in Figure 5D.

o Here it is also unclear to me, why ether/odd-chain-species for PC and PE were not included into calculations despite such species represent major species e.g. PE33:1/PE(O-)34:1 (Fig. 5S). In contrast, for PS and PI ether/odd-chain-species were included. Is there any reason for this data selection? The lipidomic approach is not able to differentiate ether and odd chain species – so I would recommend annotating species based on knowledge (odd chain species could also indicate the presence of branched chain acyl chains; please indicate when assumption are applied). In general, I would recommend to perform a separate calculation for ether species, if their presence in the worms is justified by literature. When the literature provides evidence for odd chain species they should be added into calculation for the respective lipid classes. Are these changes statistical significant

because the Fig. legend describes “There is a shift from PUFAs to saturated FA andMUFAs.”

It appears that the authors have now addressed this concern. They should only be reporting on data that can be confirmed via fragmentation profiles from the precursor.

Thank you for your assessment. Indeed, all reported lipid data is supported by mass fingerprints of precursor ions as filtered in Q1 and class-specific fragment ions as filtered in Q3. In cases when multiple interpretations are possible like in the mentioned case of ether and odd-chain glycerophospholipids, we report those multiple lipid species in the graph.

- Concerning lipid extraction: “Phase separation depends on the solvent mixtures and on the matrix of the extracted material. This needs to be optimized for each matrix.” Yes, I fully agree with this statement especially when existing protocols are changed. Therefore, I would like to ask the authors to provide data concerning on the optimization of *C. elegans* lipids. This is required because I do not agree that using only half volume of water and chloroform (compared to Bligh & Dyer and Schwudke et al.) could be considered as “minor modification”. Solvent composition is critical for effective lipid recovery.

The authors should provide literature for the modification performed with their lipid extraction (i.e., has this lipid extraction as performed by the authors been published elsewhere?). Otherwise, the authors should remove terminology that references to a previously published lipid extraction such as the Bligh- Dyer. The reviewer is correct in stating that a reduction in water and chloroform content is not a “minor modification” to a previously published Bligh-Dyer method.

The lipid extraction procedure described in this manuscript was used previously in Hannich et al., Nat. Met. 2019.

- The data contain numerous species close or below LOD – the authors did not remove such species as requested. As pointed out above this may lead to wrong interpretation. Such kind of filtering could also not be done by interested researches because it is necessary to have insight into the raw data including for example extractions blanks – this kind of data validation has to be performed by the authors to avoid misinterpretation caused by reporting of species <LOD.

The reviewer is right in that most lipidomics studies incorporate cut-off limits based on the %CVs of quality control samples. Were QCs employed in this study? If so, what was the %CV of the lower level lipid species in question based on QC data. Lipid species with higher %CVs shouldn't be included in the analysis. Alternatively, this question can be answered in the samples were analyzed in triplicate. Using triplicate samples, an analysis of the variability for lower level lipid species can be obtained.

The phospholipid analysis is based on six independent experiments with up to six technical replicates each. The results that we obtained were consistent between the individual biological replicates. No additional QC samples were run but the limit of quantification was based on the external standard curves recorded for the internal standards. Lipid amounts reported lie well within the linear range of the internal standards.

- Annotation of cardiolipin species was not changed in the suppl. table.

Lipid annotations of cardiolipin species should be recorded based on recommendations by LipidMAPS, incorporating only information in the annotation that can be confirmed by fragmentation data.

<https://www.lipidmaps.org/tools/structuredrawing/StrDraw.pl?Mode=SetupCLStrDraw>

We based the annotation of the cardiolipin species on the recommendations of the reviewer. We provide first the identity of the fatty acid fragment detected in Q3 followed by the sum of the remaining three acyl chains as calculated from the sum formula of the precursor ion (Q1), stating total carbon and double bond numbers in the acyl chains. This information is confirmed by our fragmentation data.

- I do not understand why the authors keep refusing to calculate and discuss phospholipid to sterol ratios. May be it is related to the minor fraction of sterols? One argument of the authors is that lipid changes affect only the intestine. If so, this findings could be discussed. To show and discuss such data is scientific but not misleading as suggested by the authors. Moreover, it is not true that interested reader may calculate sterol/phospholipid ratios because sterols were presented as pmol/40K worms while other lipid data are related to phosphorus content – the unit for suppl. data for Fig. 5C pmol/? is missing! Reporting of transparent lipid data is essential to support also the model presented in Fig. 7 which links membrane biophysics including cholesterol to ER structure and LD formation.

The authors' response to this concern is sufficient.

Thank you!

In summary, the response of the authors does not fit to their statement that the raised concerns were “taken seriously”. Most of the issues raised by this reviewer were not addressed appropriately. As explained above, data and method are in part unsound. Moreover, data presentation is misleading (e.g. concerning Fig. 5D or calculation of sterol/phospholipid ratios) and lacks thorough quality control.

REVIEWERS' COMMENTS

Reviewer #4 (Remarks to the Author):

All comments have been appropriately addressed. However, the authors should consider the use of quality control for future studies and the reporting of all calibrations curves in the supplemental information for completeness of the experimental design.

REVIEWERS' COMMENTS

Reviewer #4 (Remarks to the Author):

All comments have been appropriately addressed. However, the authors should consider the use of quality control for future studies and the reporting of all calibrations curves in the supplemental information for completeness of the experimental design.

We thank the reviewer for the favorable assessment. We provide the link to the calibration curves (<https://lipidomes.epfl.ch/exps/1709>).